# Positive Difference Distribution for Image Outlier Detection using Normalizing Flows and Contrastive Data

**Robert Schmier** *robert.schmier@de.bosch.com*
*Bosch Center for Artificial Intelligence, Renningen, Germany*
*University of Heidelberg, Germany*

**Ullrich Koethe** *ullrich.koethe@iwr.uni-heidelberg.de*
*University of Heidelberg, Germany*

**Christoph-Nikolas Straehle** *christoph-nikolas.straehle@de.bosch.com*
*Bosch Center for Artificial Intelligence, Renningen, Germany*

**Reviewed on OpenReview:** *https://openreview.net/forum?id=B4J4Ox7NjA*

## Abstract

Detecting test data deviating from training data is a central problem for safe and robust machine learning. Likelihoods learned by a generative model, e.g., a normalizing flow via standard log-likelihood training, perform poorly as an outlier score. We propose to use an unlabelled auxiliary dataset and a probabilistic outlier score for outlier detection. We use a self-supervised feature extractor trained on the auxiliary dataset and train a normalizing flow on the extracted features by maximizing the likelihood on in-distribution data and minimizing the likelihood on the contrastive dataset. We show that this is equivalent to learning the normalized positive difference between the in-distribution and the contrastive feature density. We conduct experiments on benchmark datasets and compare to the likelihood, the likelihood ratio and state-of-the-art anomaly detection methods.

## 1 Introduction

If an image at test time is not similar to the training images, but stems from another distribution, classical neural networks may classify the image incorrectly with high confidence(Heaven et al., 2019; Boult et al., 2019). The detection of potential anomalies, also called out-of-distribution detection or outlier detection, is a central problem in modern machine learning and is crucial for safe and trustworthy neural networks: At test time, we want to know if a given input stems from the same distribution as the training data, in order to know if we can trust our trained net on that input.

This may not significantly influence the prediction performance at test time, either because anomalous events are that rare, or because the test data is similar to the training data, but introduces a major security risk: If the camera of a self-driving car has a malfunction or something is occluding the view, the car should be able to detect the situation as anomalous and should not use the input of the camera. The use of outlier detection to improve performance for rare events or unseen data is manifold: One can use outlier detection while training to purify the data set (Zhao et al., 2019b), at train time to give rare training data points a stronger training signal (Steininger et al., 2021), or at test time to find and react to anomalies (Wang et al., 2021). In contrast to discriminative networks, where the texture (Geirhos et al., 2020) or the background (Beery et al., 2018) can be sufficient to get a satisfactory performance, for outlier detection the network must distinguish between the in-distribution data and all other -unknown- data. The network therefore must "understand" what the in-distribution data characterizes and following Richard Feynman's famous saying "What I cannot create, I do not understand", we believe that generative models are therefore the most promising approach to outlier detection. We turn to the fast growing field of normalizing flows (Kobyzev et al., 2020), which

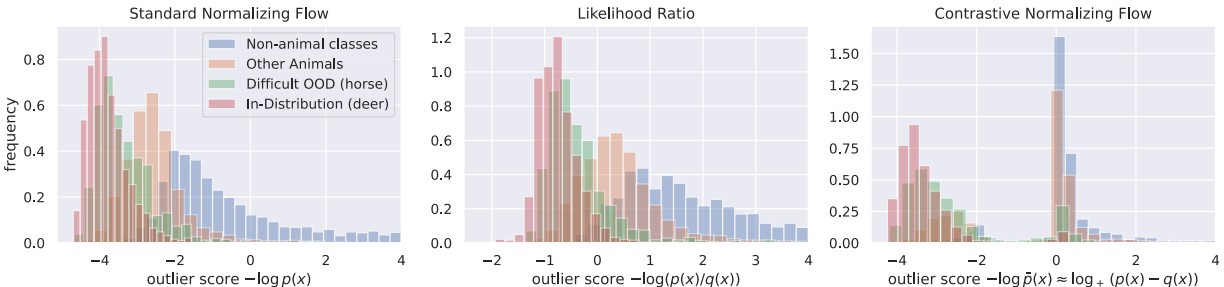

Figure 1: Histogram of the outlier scores of all CIFAR-10 classes under models trained with the inlier class deer. The semantically similar classes deer and horse are difficult to separate for all methods, our method "contrastive normalizing flow" (right) achieves good separation for the other classes. We group the other CIFAR-10 classes for this figure into an animal (frog, cat, dog and bird) and a non-animal super-class (truck, car, plane and ship) for better readability. We abbreviate $\log_+(\cdot)$ as the logarithm of the positive difference for the outlier score of the contrastive normalizing flow.

allow exact density estimation, fast sampling, and suffer (almost) no mode collapse. While an important line of research deals with detecting small anomalies within an image, so-called defects, in industrial settings (e.g., Roth et al. (2022)), we focus in this work on **semantic single-shot outlier detection:**

**Task** The goal is to find outliers on image level: An outlier is an image of a different semantic class, which is not known at training time. We train e.g., solely on images of the class deer of the CIFAR-10 dataset and want to distinguish between images of the in-distribution class deer and the other CIFAR-10 classes at test time. In contrast to distribution-based outlier detection where a set of many test-samples/measurements is compared to the training distribution (see e.g. Coluccia et al. (2013); Bouyeddou et al. (2018)) we focus on the single-shot setting, where the inlier vs. outlier decision must be made independently for individual data points. Finding semantic outlier images is a common task definition in the outlier detection literature, e.g., Ren et al. (2019); Schirrmeister et al. (2020); Nalisnick et al. (2019b;a); Kirichenko et al. (2020); Choi et al. (2018).

**Approach** Inspired by the work of Ren et al. (2019) and Schirrmeister et al. (2020), who used the ratio of likelihoods for outlier detection, we modify the training objective for normalizing flows to learn the positive *difference* of two distributions: An inlier distribution $p$ and an contrastive distribution $q$. We prove that this is achieved by maximizing the log-likelihood on data which stems from $p$ while minimizing the log-likelihood on data stemming from $q$. We use the negative log-likelihood under this usually intractable difference distribution given by our normalizing flow as outlier score and demonstrate that this new score is especially suited for outlier detection. We argue and demonstrate experimentally that the difference $p - q$ is much more robust than the ratio $p/q$: In regions where $p$ and $q$ are small, there are only few training instances, and the estimators of $p$ and $q$ will accordingly be very noisy. The quotient between two noisy small numbers can become very big, whereas their difference remains small.

Moreover, we run experiments comparing different contrastive datasets and show that our method is robust under the choice of the contrastive distribution. We show that our method works well with an contrastive distribution broader than the inlier distribution, and it does not harm the performance if the contrastive distribution contains inlier data. We do not use label information, but see the contrastive dataset as a collection of unlabeled images. We motivate the use of such an auxiliary dataset twofold: Collecting unlabeled images is easy and cheap, but labeling them would be expensive. Since applying normalizing flows on images is computationally intensive, we do not train directly on images but employ MoCo He et al. (2020), a dimensionality reducing feature extractor trained in a self-supervised manner on IMAGENET (Deng et al., 2009). This fits the ongoing development of employing pretrained models to exploit prior knowledge (Bergman et al., 2020). A schematic overview of our method is shown in figure 2. To give an introductory example, we compare the outlier scores for all CIFAR-10 classes given by a standard normalizing flow, the ratio method

and our proposed method trained on the deer class of the CIFAR-10 dataset as in-distribution data and IMAGENET as contrastive dataset in figure 1. Our method separates semantically distant classes better than existing methods.

We focus on normalizing flows as density estimators, since they are universal approximators with stable training procedure. However, the new objective and theory is applicable to any density estimator trainable with explicit maximum likelihood, as is required for our new objective in equation 3. This includes Mixture-of-Gaussian density estimators or binning methods in lower dimensional settings.

Our method "Positive Difference Distribution for Image Outlier Detection using Normalizing Flows and Contrastive Data" combines a generative model with exact density estimation (normalizing flow) and a new training objective for density estimators. We employ a pre-trained feature extractor for further performance improvement. Our contributions are:

- A new likelihood-based outlier score

- A new objective to train density estimators

- Theoretical results relating our new objective to negative log-likelihood training of an intractable difference distribution

- Experimental validation of the theoretical advantages compared to the likelihood-ratio method

- State-of-the-art results in experiments on standard semantic outlier detection benchmarks

## 2 Related work

**Outlier detection**  Outlier detection, also known as out-of-distribution or anomaly detection, is the task of detecting unknown images at test time. See Salehi et al. (2021) for a extensive discussion of the field. In this work, we will focus on semantic outlier detection, by either using one class of a given dataset as inliers and all other classes as anomalies (one-vs-rest setting) or by using a complete dataset as inliers and comparing against other datasets (dataset-vs-dataset setting).

Most related work can be categorized as either reconstruction based or using a representation ansatz: In a reconstruction approach, the model tries to reconstruct a given image, and the score relies on the difference between the reconstruction and the original image. This idea goes back to Japkowicz et al. (1995) and has been applied to various domains, e.g. time series (Zhang et al., 2019), medical diagnosis (Lu and Xu, 2018) and flight data (Memarzadeh et al., 2020). Recent models used for image data are reconstruction with a memory module in the latent space (Gong et al., 2019) or combinations of VAE and GAN (Perera et al., 2019). An interesting new idea is the combination of VAE and energy-based models by Yoon et al. (2021), where the reconstruction loss is interpreted as the energy of the model. For the representation approach the model tries to learn a feature space representation and introduces a measure for the outlierness in this representation: Ruff et al. (2018) map all data inside a hypersphere and define the score as the distance to the center. Another approach uses transformations to either define negatives for contrastive learning (Tack et al., 2020) or to directly train a classifier (Bergman and Hoshen, 2020). Zong et al. (2018) fit a gaussian mixture model to the latent space of an autoencoder.

**Normalizing flows**  While most generative models are either able to easily generate new samples (like GANs or VAEs) or give an exact (possible unnormalized) density estimation (like energy-based models), Normalizing flows, first introduced by Rezende and Mohamed (2015) are able to perform both tasks. Normalizing flows rely on the change of variables formula to compute the likelihood of the the data and therefore need a tractable Jacobian determinant: Most normalizing flows achieve this by either using mathematical "tricks" (e.g., Rezende and Mohamed (2015)), as autoregressive models which restrict the Jacobian to a triangular shape (e.g., Kingma et al. (2016)) or by special architectures which allow invertibility (e.g., RealNVP by Dinh et al. (2017)). RealNVP makes use of coupling blocks to apply an affine transformation on a subset of the features conditioned on the subset's complement.

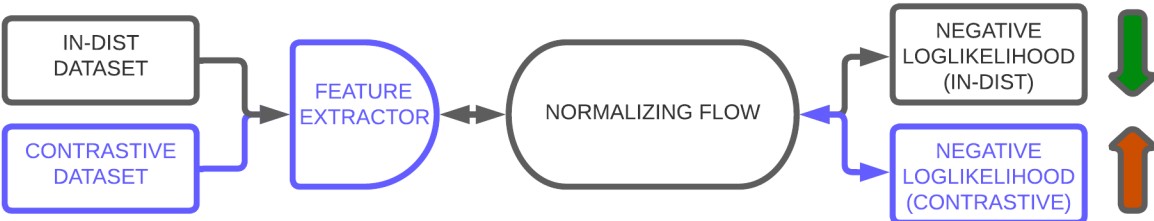

Figure 2: Schematic overview of our method. Standard normalizing flow in black. The feature extractor is an optional extension, which further improves the performance for image data. We do not use an feature extractor for tabular data and for our toy experiments.

**Density of normalizing flows for outlier detection** The use of generative models seems like a perfect fit for outlier detection: While Classifiers tend to focus on features to distinguish the given classes, generative models need to include all relevant information to be able to generate new samples. Methods with exact density estimation (e.g., normalizing flows, energy-based methods) directly offer a good outlier score via the learned density. Unfortunately, normalizing flows work poorly in the case of outlier detection when employed directly on images: Various works showed that the likelihood is dominated by low-level statistics (Nalisnick et al., 2019a) and pixel-correlations (Kirichenko et al., 2020) and fail to detect anomalies (see also Zhang et al. (2021)). There have been multiple attempts to improve the outlier score directly by introducing a complexity measure (Serrà et al., 2020), using typicality (Nalisnick et al., 2019b), using hierarchies (Schirrmeister et al., 2020) or employing ensembles (Choi et al., 2018).

**Likelihood ratio for outlier detection** Another line of research is using the ratio of two likelihood models $p/q$ as outlier score: Ren et al. (2019) use an augmented version of the in-distribution as contrastive distribution and computed the likelihood ratio with Pixel-CNN, an autoregressive generative model. Schirrmeister et al. (2020) train separate normalizing flows on the in-distribution data and Tiny Images as contrastive dataset. They require the in-distribution dataset to be included in the contrastive dataset to meet their hierarchical principle.

**Feature extractor** With the broad availability of trained deep models and their good generalization capacity, the use of such models as feature extractor has gained popularity: Most implementations use the output of intermediate layers of a model trained as a discriminator on an auxiliary dataset (e.g. IMAGENET by Deng et al. (2009)) as features: Prominent examples are ResNet (He et al., 2016), used by e.g., Cohen and Hoshen (2020) and Roth et al. (2022), or ViT (Dosovitskiy et al., 2021), used by Cohen and Avidan (2022) for outlier detection. We use a feature extractor trained in a self-supervised fashion via a contrastive learning objective (Chen et al., 2020). Their and the improved method by He et al. (2020) showed remarkable results on down-stream tasks without relying on label information.

**Combining feature extractors and normalizing flows** An existing line of work uses the representations given by pretrained feature extractors to detect outliers: Cohen and Hoshen (2020) use the intermediate layer of a pretrained ResNet as the feature representation and use the sum of distances to the k-nearest Neighbours as their outlier score. Yu et al. (2021) and Rudolph et al. (2021) train a unconditioned normalizing flow on a feature space of a pretrained feature extractor trained on IMAGENET, while Gudovskiy et al. (2022) use conditional normalizing flows (Ardizzone et al., 2019). All of these methods focus mainly on defect detection and localization, while our paper works on the task of semantic outlier or anomaly detection. In contrast to MoCo (He et al., 2020), which is used in this work for the feature extraction, all their feature extractors are trained in a supervised fashion.

## 3 Background

### 3.1 Normalizing flows

Normalizing flows are a class of generative models, which allow exact density estimation and fast sampling. A comprehensive guide to normalizing flows can be found in Papamakarios et al. (2021b). A normalizing flow maps the given data via an invertible transformation $T^{-1}$ to a chosen base distribution of the same dimensionality, e.g., a multivariate normal distribution by minimizing the empirical KL-Divergence between the transformed data distribution in the latent space and the base distribution. This is equivalent to minimizing the negative log-likelihood of the training data under the model. This likelihood can be calculated by the change of variable formula. Therefore, the Jacobian of the transformation needs to be tractable. For the normal distribution as base distribution, the likelihood can be computed:

$$L(\theta) = - \sum_{\boldsymbol{x} \in X} \log p_\theta(\boldsymbol{x}) = \sum_{\boldsymbol{x} \in X} \frac{||T_\theta^{-1}(\boldsymbol{x})||_2^2}{2} - \log |\mathrm{Jac}_{T_\theta}(\boldsymbol{x})| + C, \tag{1}$$

where $X$ is the training data, $\theta$ are the parameters of the invertible transformation $T$, $\mathrm{Jac}_T$ denotes the determinant of the Jacobian and $C$ is a constant independent of $\theta$. To apply the change of variable formula in equation 1 the transformation has to be invertible and needs a tractable determinant of the Jacobian. We achieve this by using the realNVP architecture established by Dinh et al. (2017).

### 3.2 Application of a feature extractor and MoCo

Standard normalizing flows, e.g., Ardizzone et al. (2019); Dinh et al. (2017); Kingma and Dhariwal (2018), are directly applied on the data space to allow exact density estimating and sampling of new data following the in-distribution. As we are interested in outlier detection and do not need new samples generated by the model, we follow a different approach: We employ our density estimation on a pretrained feature space. The outlier detection capability of a normalizing flow is improved in this case, as demonstrated by Kirichenko et al. (2020). We verify this by running experiments without the use of a feature extractor in the appendix for standard outlier methods (section A.3) and flow methods (section A.9).

Additionally, using a feature extractor leads to lower computational costs: While for a GLOW normalizing flow (Kingma and Dhariwal, 2018) [1] a simple forward task with a single IMAGENET sample takes 162 ms $\pm$ 1.91 ms, our method (a frozen MoCo feature extractor and a Freia (Ardizzone et al., 2018-2023) AllInOne normalizing flow in 128 dimensions) takes only 11.4 ms $\pm$ 0.25 ms. This difference increases for a training loop (ours 15.5 ms $\pm$ 0.721 ms versus GLOW 338 ms $\pm$ 2.45 ms).

We use a pretrained MoCo (He et al., 2020) feature extractor, which has been trained with a self-supervised objective on IMAGENET. Self-supervised contrastive methods learn a representation by maximizing the similarity of different views of the same image in the feature space. For a training step, every training image is augmented twice, by augmentations consisting of color distortions, horizontal flipping, and random cropping. These augmented images are fed into an encoder network and for a positive pair $\boldsymbol{z}_0, \boldsymbol{z}_1$ and negative representations $\boldsymbol{z}_2, ..., \boldsymbol{z}_N$ (other augmented images) the infoNCE loss is optimized for all augmented images:

$$l_0 = - \log \frac{\exp(sim(\boldsymbol{z}_0, \boldsymbol{z}_1)/\tau)}{\sum_{k=1}^N \exp(sim(\boldsymbol{z}_0, \boldsymbol{z}_i)/\tau)}. \tag{2}$$

$\tau$ is a temperature scalar and $sim(\boldsymbol{x}, \boldsymbol{y})$ denotes the cosine similarity (cosine of the angle) between $\boldsymbol{x}$ and $\boldsymbol{y}$. Oord et al. (2018) proved that optimizing this infoNCE loss is maximizing a lower bound to the mutual information between the input data and the learned representation. As we are interested in semantic anomalies of the data distribution, this maximal informative representation is well suited for our method. While MoCo is trained without any label information, one could also use e.g., a classifier trained on a labeled auxiliary dataset as feature extractor. To show the advantage of the MoCo feature extrator over a supervised feature extractor trained without the infoNCE, we run an ablation experiment on inceptionV3 and report the results in the appendix in A.4.

---

[1]We use the GLOW pytorch implementation of `https://github.com/rosinality/glow-pytorch` and used the default hyperparameters.

## 4  Method: Contrastive normalizing flow

We introduce our method "contrastive normalizing flow" as a novel way to train normalizing flows: We adapt the training of normalizing flows by maximizing the log-likelihood on in-distribution data $\boldsymbol{x} \sim p$ and minimizing the log-likelihood on a broader contrastive dataset $\boldsymbol{y} \sim q$. The intuition behind this approach is simple: We want the model to learn high likelihoods on in-distribution data and low likelihoods everywhere else. We use the negative log likelihood given by the model trained with this novel training objective as outlier score. We train our model with the following objective:

$$L = \min_{\theta} \mathbb{E}_{\boldsymbol{x} \sim p}[-\log p_{\theta}(\boldsymbol{x})] - \mathbb{E}_{\boldsymbol{y} \sim q}[-\log p_{\theta}(\boldsymbol{y})]. \tag{3}$$

Rearranging the loss shows that instead of learning p, we learn a truncated version of $p - q$ to the region where $p > q$:

$$L = \min_{\theta} - \int \log p_{\theta}(\boldsymbol{x})[p(\boldsymbol{x}) - q(\boldsymbol{x})]d\boldsymbol{x} \tag{4}$$

$$= \min_{\theta} - \int_{\text{supp}(p>q)} \log p_{\theta}(\boldsymbol{x})[p(\boldsymbol{x}) - q(\boldsymbol{x})]d\boldsymbol{x} + \min_{\theta} \int_{\text{supp}(q>p)} \log p_{\theta}(\boldsymbol{x})[q(\boldsymbol{x}) - p(\boldsymbol{x})]d\boldsymbol{x} \tag{5}$$

$$= \frac{1}{C} \min_{\theta} \mathbb{E}_{\boldsymbol{x} \sim C(p-q)1_{\{p>q\}}}[-\log p_{\theta}(\boldsymbol{x})] + \min_{\theta} \int_{\text{supp}(q>p)} \log p_{\theta}(\boldsymbol{x})[q(\boldsymbol{x}) - p(\boldsymbol{x})]d\boldsymbol{x}, . \tag{6}$$

In the second term, $q > p$. In this case, L is minimized by $p_{\theta} \to 0$. In words: The learnt probability tries to put no mass to the learned distributions in regions where there is more contrastive data than in-distribution data.

The first term in equation 6 is the negative log-likelihood objective for the normalized positive difference distribution $\bar{p} = C(p - q)1_{\{p>q\}}$, with new normalizing constant $C$. This term is maximized by $\hat{\theta}$ with $p_{\hat{\theta}}(\boldsymbol{x}) = C(p - q) \ \forall \boldsymbol{x} \in \text{supp}(p > q)$ (Papamakarios et al., 2021a).

By training our model with equation 3, we learn the normalized positive difference between in-distribution and contrastive distribution. This is in contrast to a standard normalizing flow, where we learn the distribution of the in-distribution samples. We use the negative logarithm of this new - otherwise intractable - difference density as outlier score. The pseudocode for our training procedure is given in the appendix in section A.1. We clamp $\log p_{\theta}(x)$ on the contrastive dataset during training by a threshold $\epsilon$ in eq 1, see appendix A.2.1 for a discussion.

When separating the probability density into a semantic part and a low-level pixel correlation part, Ren et al. (2019) argue that the common low-level correlations are the same for all natural images. This strong assumption is not necessary for our theory to hold but can motivate the method and give the reader some intuition. Using this reasoning, we separate the feature dimensions of our data in a high level semantic representation $\boldsymbol{s}(\boldsymbol{x})$ and a low-level feature representation $\boldsymbol{f}(\boldsymbol{x})$, where the low level pixel feature conditional distribution $p(f|s)$ is the same for all natural images and $(\boldsymbol{f}, \boldsymbol{s})$ are a volume preserving transformation such that $p(\boldsymbol{x}) = p(\boldsymbol{f}(\boldsymbol{x}), \boldsymbol{s}(\boldsymbol{x}))$. We can now rewrite the difference between the in-distribution $p$ and the contrastive distribution $q$ as

$$p(\boldsymbol{x}) - q(\boldsymbol{x}) = p(\boldsymbol{f}|\boldsymbol{s})p(\boldsymbol{s}) - q(\boldsymbol{f}|\boldsymbol{s})q(\boldsymbol{s}) = p(\boldsymbol{f}|\boldsymbol{s})(p(\boldsymbol{s}) - q(\boldsymbol{s})).$$

This results in a low score when either the low-level features have a low density given the semantic content of the image or the semantic likelihood of the image is higher for the broader contrastive distribution than for the inlier distribution. We find both cases important classes of anomalies.

**Outlier score**  The outlier score is the *negative* log density given by our model, which is equivalent to the negative log-likelihood of the positive difference density described in equation 6. By setting a threshold $\delta$, all inputs with a outlier score greater than $\delta$ are classified as anomalies. For evaluating, we use area under the receiver operating characteristic (AUROC).

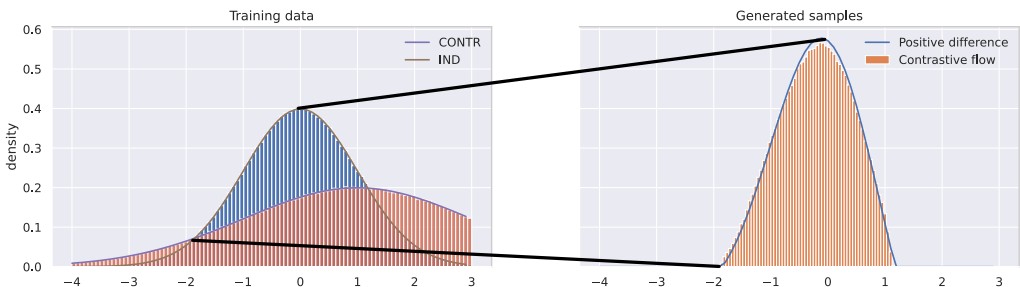

Figure 3: Example of an 1D contrastive normalizing flow: We train on samples $x \sim p = \mathcal{N}(0,1)$ as in-distribution and $y \sim q = N(1,2)$ as contrastive dataset. The training distributions $p$ and $q$ are shown on the left. The learned distribution of our contrastive normalizing flow method and the analytic normalised positive difference between the ground truth densities are shown on the right. The black lines connect equivalent points in both diagrams.

Table 1: Comparison of our contrastive normalizing flow with the standard normalizing flow and the ratio method.

| Method | Normalized score | Single model | Use of contrastive data | Behaviour for p,q«1 | Reduces influence of low-level features |
|---|---|---|---|---|---|
| Standard NF | ✓ | ✓ | ✗ | stable for p«1 | ✗ |
| Ratio method | ✗ | ✗ | ✓ | unstable | ✓ |
| Contrastive NF | ✓ | ✓ | ✓ | stable | ✓ |

**1D Toy example**  We conduct a qualitative 1D experiment to demonstrate that our derived theory holds by training a contrastive normalizing flow on samples $x \sim p = \mathcal{N}(0,1)$ as in-distribution and $y \sim q = \mathcal{N}(1,2)$ as contrastive dataset. We show in figure 3, that the density learned by the model exactly matches the positive difference density as described in equation 6. We do an ablation study on the role of the clamping-hyperparameter $\epsilon$ in the appendix A.4.

**Outlier detection and comparison to ratio method**  We argue that our learned difference distribution is especially suited for outlier detection: When using a broader distribution as contrastive distribution we can assume that $p(\boldsymbol{x}) > q(\boldsymbol{x})$ holds for in-distribution data. In regions where the contrastive distribution $q(\boldsymbol{x})$ has a high density the learned positive difference will be small or zero. This behaviour is similar to the likelihood ratio method of Ren et al. (2019): They train two density estimators, one on the inlier distribution, the other one on the contrastive distribution and use their ratio as outlier score. An advantage of our method compared to the likelihood ratio is that our learned density $p_\theta(\boldsymbol{x})$ is robust in areas where $p(\boldsymbol{x})$ and $q(\boldsymbol{x})$ are very small, i.e., where the in-distribution data and the contrastive distribution data are sparse. The learned density $p_\theta$ is normalized and therefore integrates to 1. Thus, in the areas where both distributions $p$ and $q$ have no support the difference density is zero or almost zero. When comparing $-\log p_\theta(\boldsymbol{x})$ at test time to a threshold to see whether it is an inlier (less than threshold) or outlier (greater than threshold) data points outside of the support of $p$ and $q$ will be reliably detected as outliers. In contrast, the density ratio $p(\boldsymbol{x})/q(\boldsymbol{x})$ is ill-defined in regions where both $p(\boldsymbol{x})$ and $q(\boldsymbol{x})$ have little or no support. The division of two small numbers can either be very large or very small. This leads to problems demonstrated by the experiments in section 5, because the data which one might want to detect as outliers can be very far from the contrastive distribution which was used during training – the likelihood method fails due to spurious high ratios in the noisy tails of the learned distribution. We show in a 2D toy experiment, that the likelihood ratio has the highest inlier score far away from the inlier and contrastive distribution while our model is able to capture the in-distribution well. We show in table 1 an overview of the properties of our method in comparison to the ratio method and the standard normalizing flow.

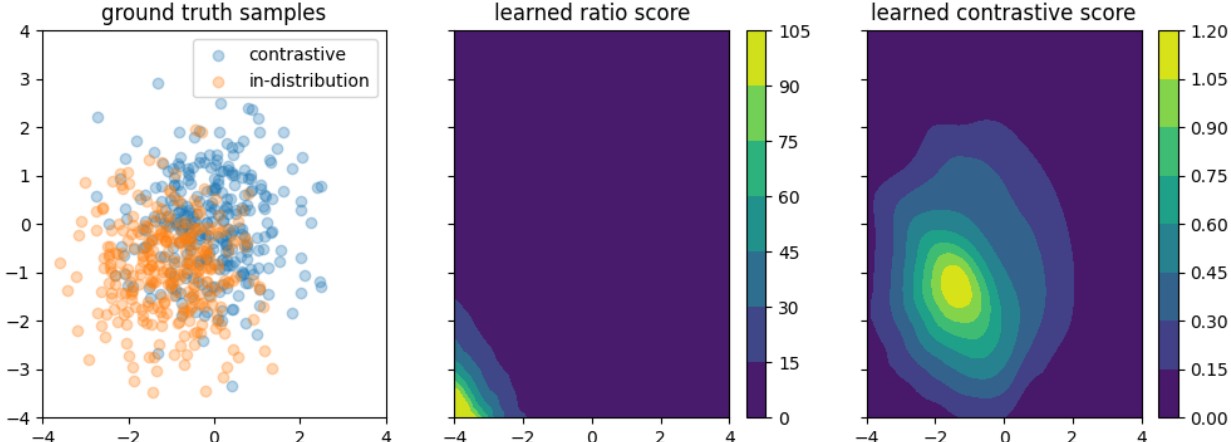

Figure 4: Learned in-distribution scores for the density ratio method and our contrastive normalizing flow on a 2D toy examples. For better visualisation, we plot the exponential of the in-distribution score (equivalent to the exponential of the negative outlier score used throughout the paper). On the left the ground truth samples for the in-distribution and the contrastive data are shown. While the ratio method has the highest in-distribution scores far away from both distributions in the lower left, the contrastive flow in-distribution score captures the inliers well.

**2D Toy Example**   To show the theoretical advantage of the learned difference density over the density ratio method, we conduct a 2D toy experiment: We learn a contrastive normalizing flow on a normal distribution centered at $[1,1]$ as in-distribution and a normal distribution centered on the origin as contrastive distribution. For the ratio method, we train a separate normalizing flow on each distribution, and take the ratio at inference time. As we see in figure 4, our contrastive normalizing captures the in-distribution data, while the ratio method has the highest in-distribution scores far from both distributions. This is not surprising, as the analytic log ratio of the two Gaussian distributions goes to infinity for $x \to [-\infty, -\infty]$.

We also show this behaviour in a real world setting in section 5.2.2.

## 5   Experiments

**Experimental outline**   First we investigate the performance of our new difference method for different choices of the contrastive dataset and compare to the likelihood ratios method. We study the case of inliers in the contrastive dataset, the behavior for dissimilar in-distribution and contrastive datasets and the supervised setting, where the contrastive dataset contains some examples of test time outliers. To compare our method to other semantic outlier detection methods, we conduct experiments on CIFAR-10, CIFAR-100 and CelebA.

### 5.1   Implementation details

To extract useful features, we use a MoCo encoder pretrained on IMAGENET (He et al., 2020). We show the inferior performance for another feature extractor in the appendix in section A.4. As features we use the output of the network, this is in contrast to the original MoCo implementation, where classification is done via an MLP head on the penultimate layer. Working with the last layer results in a better performance for outlier detection. Because the contrastive MoCo objective is invariant under changes of the norm of the features, we normalize all features to a hypersphere and add small noise afterwards to obtain a non-degenerated density, we discuss the normalization in the appendix in section A.4. For the normalizing flow implementation, we use the "Framework for Easily Invertible Architectures" (Ardizzone et al., 2018-2023). Unless otherwise specified, we use eight of their "AllinOne"-blocks for our architecture. We list all hyperparameter for training in section A.2.3 in the appendix. As mentioned in the method section 4, we

clamp the contrastive likelihood to prevent the loss from diverging. We discuss the clamping in the appendix in subsection A.2.1. We finetuned the pretrained MoCo feature as discussed in the appendix in A.2.2.

## 5.2 Investigation of the contrastive distribution and comparison to ratio method

To investigate the role of the contrastive distribution and to get insights on the difference between our method and the ratio method proposed by Ren et al. (2019), we run ablation studies on different dataset mixtures as contrastive distribution. We run most of our ablation experiments on the deer and the horse class, as our model and the baselines struggle to distinguish these two classes (see table 3 for the full confusion matrix), and experiments are most informative here.

### 5.2.1 Contrastive distribution containing in-distribution data

In this ablation experiment the performance for a contrastive distribution containing in-distribution samples is measured. When including inliers in the contrastive dataset, we expect to see no change in performance following our theoretical derivation: With in-distribution data $p$ and pure contrastive data $q$ (and learned positive difference $\bar{p} = \bar{C}(p - q)1_{p>q}$), we get for the mixed contrastive data $q_{new} = (1 - \mu)p + \mu q$:

$$\bar{p}_{new} = C(p - q_{new})1_{p>q_{new}} \tag{7}$$
$$= C(p - (1 - \mu)p - \mu q)1_{p>(1-\mu)p+\mu q} \tag{8}$$
$$= C(\mu p - \mu q)1_{\mu p>\mu q} \tag{9}$$
$$= C\mu(p - q)1_{p>q} \tag{10}$$
$$= \bar{p}, \tag{11}$$

with $C\mu = \bar{C}$. To show this qualitatively, we use the horse class of the CIFAR-10 dataset as inlier class and a mixture of IMAGENET and in-distribution data as contrastive dataset. We denote the mixture fraction of IMAGENET for the contrastive dataset as $\mu = \frac{\text{number of IMAGENET samples}}{\text{contrastive dataset size}}$, so for $\mu = 1$ the contrastive dataset consists only of imagenet samples and for $\mu = 0$ the contrastive dataset contains only in-distribution data. We report the AUROC scores for different mixing fractions $\mu$ in figure 5. One can see that the performance of our method stays constant for $\mu > 0$ and reliably beats the standard normalizing flow, where the ratio method drops in performance as $\mu$ approaches zero and the contrastive distribution contains more and more inliers. For $\mu = 0$, as in-distribution and contrastive distribution coincide, both models do not get a useful training or detection signal and outlier detection fails for both (AUROC score approx. 50).

### 5.2.2 Use of a narrow contrastive distribution

In this ablation study, we want to investigate the performance for a contrastive dataset with a small semantic variety, e.g., a single semantic class. We call such a contrastive dataset narrow. We use the deer class of CIFAR-10 as in-distribution data and a mixture of IMAGENET and the horse class of CIFAR-10 as contrastive dataset. We denote the mixture fraction of IMAGENET as $\mu = \frac{\text{number of IMAGENET samples}}{\text{dataset size}}$, so for $\mu = 1$ the contrastive dataset consists only of imagenet samples and for $\mu = 0$ the contrastive dataset contains only samples from the CIFAR-10 horse class. We plot in figure 6 the AUROC scores for the horse class included in the mixture and the other CIFAR-10 classes. While the performance of the ratio method drops significantly (AUROC 96.5 to 75.6) when using a single narrow CIFAR-10 class as contrastive dataset ($\mu = 0$), our method only drops slightly (AUROC of 96.7 to 93). We plot in figure 7 the histograms for in-distribution, contrastive and out-of-distribution data for only the horse class as contrastive dataset. We see that the ratio method fails and assigns a lower outlier score to out-of-distribution examples than to in-distribution data, especially for the setting of no IMAGENET samples in the contrastive distribution. Our difference method stays almost constant for reasonable mixtures of IMAGENET and the contrastive class. For a small mixture fraction $\mu$, our model is outperformed by the standard normalizing flow by a small margin because of the overlap of in-distribution and contrastive class in the MoCo feature space. We conclude that our method is more robust than the ratio method for outliers far from the inlier and contrastive distribution as discussed in section 4 and the results of the toy experiment in section 4 transfer to real world data.

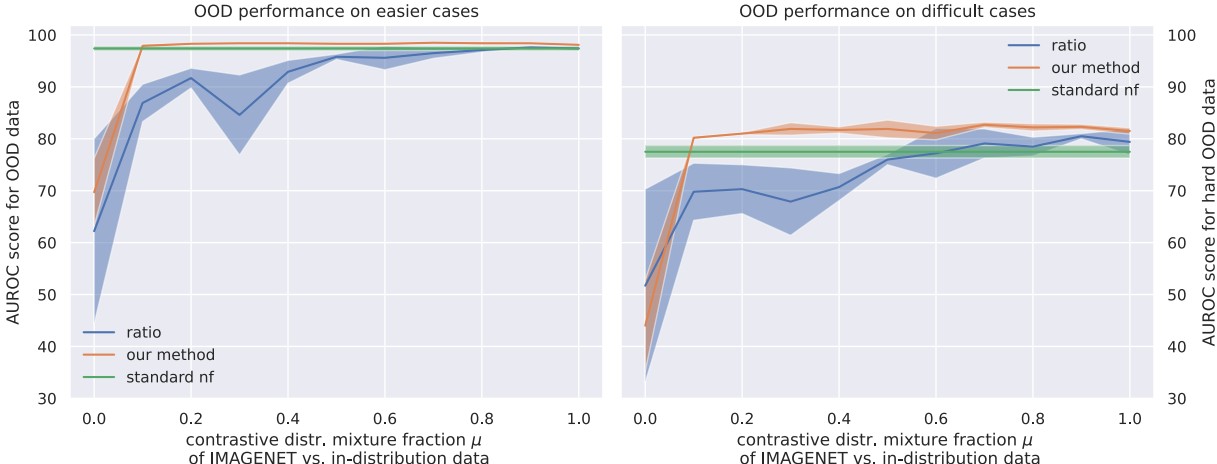

Figure 5: Performance for our model, a standard normalizing flow and the ratio method trained on the CIFAR-10 horse class as in-distribution and of a mixture of IMAGENET and in-distribution data as contrastive dataset. $\mu$ denotes the fraction of imagenet samples in the contrastive dataset, so for $\mu = 1$ the contrastive dataset consists only of imagenet samples and for $\mu = 0$ the contrastive dataset contains only samples from the CIFAR-10 horse class. On the right we plot the AUROC score against difficult out-of-distribution cases (deer class of CIFAR-10), on the left against all other CIFAR-10 classes (without horse and deer). Errors are computed over three different seeds.

### 5.2.3 Use of test time anomalies at training time

In this ablation study, the performance in a partially supervised setting is examined, where the contrastive data is a mixture of a broad data distribution and anomalous data. We use the same setting as in the previous experiment: the deer class of CIFAR-10 as inlier and a mixture of IMAGENET and the horse class of CIFAR-10 as contrastive dataset. We measure the outlier performance between the inlier class and the horse class - this is in contrast to all other outlier detection experiments in the paper, as we explicitly use splits of the same distribution as contrastive data and training time and anomalous data at test time. We show the performance for different mixture fractions of our method, a ratio method and a normalizing flow as comparison in figure 6. We see that the performance of our model increases already with a small percentage of anomalous data in the mixture. Both methods saturate at an AUROC score of about 95, we conclude that both classes have a significant overlap in the MoCo feature space and perfect separation is even in the fully supervised case not possible.

### 5.2.4 Findings

Using the results of the previous experiments, we formulate the following findings and use them to select a suitable contrastive distribution for the benchmark experiments:

**Difference method improves on previous flow methods** In all conducted experiments our difference method is on par or outperforms the ratio method and improves on standard normalizing flows for broad contrastive distributions or when the test-time outlier distribution is known at training time. Especially for narrow contrastive distributions, our difference model shows a significant improvement. When contaminating the contrastive data with in-distribution data, our method is able to use a small fraction of unknown contrastive datapoints for a stable performance, where the ratio method shows a strong performance degradation. Additionally, our model has another interesting advantage: While both methods can use the same architecture, our difference method only needs one model at test time, whereas the ratio method needs separate in-distribution and contrastive models.

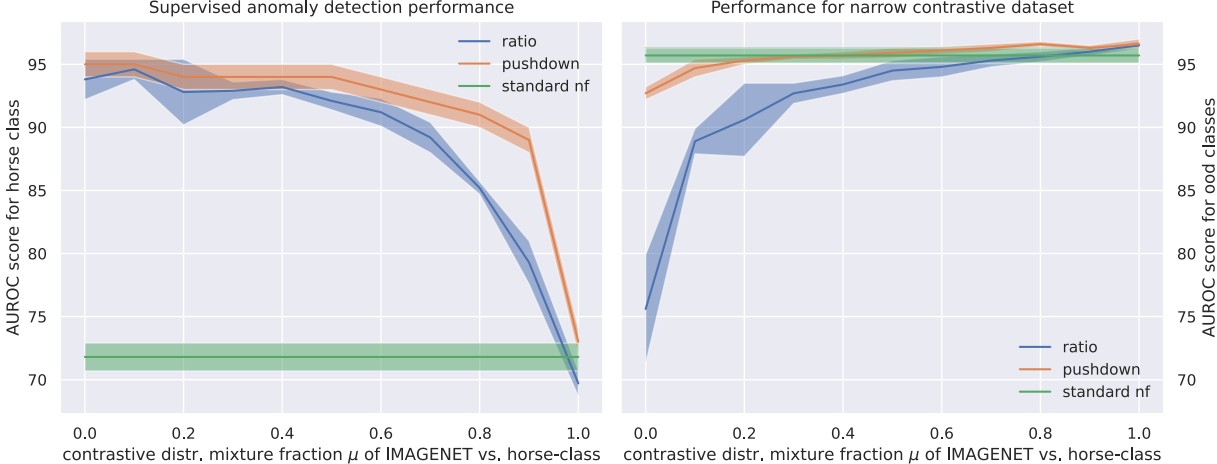

Figure 6: Performance for our model, the ratio method and a standard normalizing flow trained on the CIFAR-10 deer class as in-distribution and a mixture of IMAGENET and the horse class of CIFAR-10 as contrastive dataset. On the left we plot the AUROC score against the horse class included in the mixture (which correspond to a fully supervised setting when $\mu = 0.0$), on the right against all other CIFAR-10 classes (without horse and deer). For a contrastive dataset containing only images of the CIFAR-10 horse class ($\mu = 0.0$), the ratio methods fails, when our method only slightly drops in performance. Standard deviations are computed over three different seeds. Please note that although both graphs use the same trained model, the experimental setting is vastly different: On the left the distribution of the outliers is already known at training time, where on the right we stay in the uniformed outlier detection setting.

**Do not use narrow contrastive distributions for uniformed outlier detection**  When choosing a narrow dataset (e.g., a single other semantic class) as contrastive distribution, the performance of our method and the ratio method drops significantly when the contrastive samples differ from the test time outliers ($\mu = 0.0$ in the right plot of figure 6). In this setting, both methods are outperformed by a standard normalizing flow trained without the use of a contrastive dataset.

**How to choose a good contrastive dataset**  The performance of our method neither improves nor declines when including inlier data in the contrastive dataset. Even for contrastive datasets consisting of ninety percent inlier data, the performance of our difference method stays almost constant. In the uninformed outlier detection setting, a broad contrastive distribution works best, even if it contains some inlier data. Including known outliers in the contrastive distribution helps in the informed outlier detection setting. However, only using a set of narrow known outliers as contrastive data would hamper the performance on other unknown outliers.

**Contrastive distribution for benchmark experiments**  Following our findings, we use IMAGENET as a broad contrastive dataset for our benchmark experiments on CIFAR-10, CIFAR-100 and CelebA. IM-AGENET is the same dataset used in the training of the MoCo feature extractor.

We use the same experimental setup but without the use of a feature extrator to train directly on images to disentangle the effect of the feature extractor and the contrastive normalizing flow and report the results in the appendix in section A.9. We see a similar behaviour for the dependence of the outlier detection performance on the mixture fraction $\mu$ with and without the use of a feature extractor, but a performance decrease in uninformed outlier detection performance without a feature extractor. The results support our findings in this section and confirm our choice of employing a feature extractor before applying the contrastive normalizing flow.

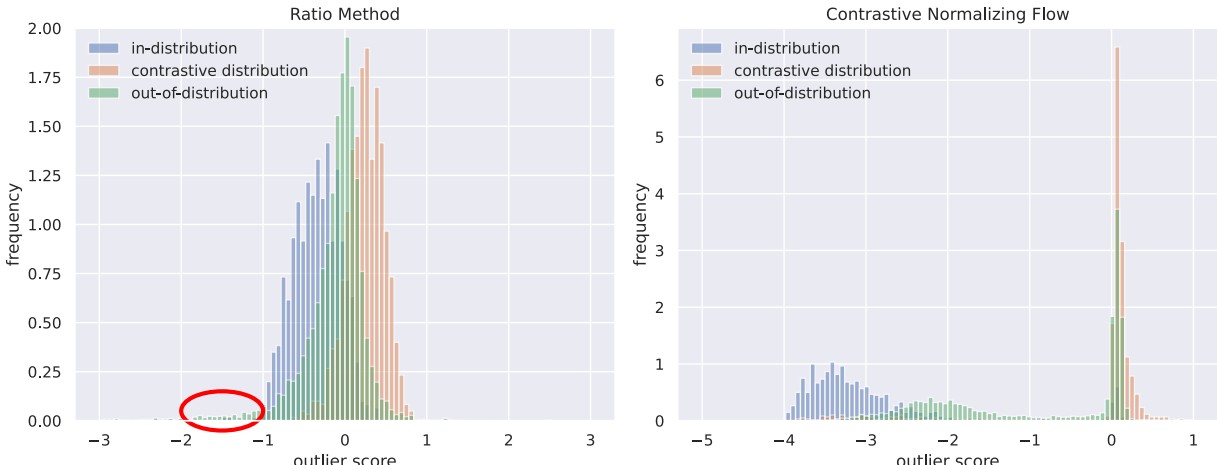

Figure 7: Histograms of the outlier scores for in-distribution, contrastive and ood data for our model (right) and the ratio method (left) trained on the CIFAR-10 deer class as in-distribution and the horse class of CIFAR-10 as contrastive dataset. The out-of-distribution data are all other CIFAR-10 classes. The models coincide with the models for $\mu = 0.0$ in figure 6. While for the contrastive flow in-distribution samples have the lowest outlier scores, the tail of the out-of-distribution data (highlighted with a red circle in the figure) surpasses the in-distribution data for the ratio method and out-of-distribution samples have the lowest outlier score.

## 5.3 Benchmark experiments

**Our methods** We run experiments for two versions of our contrastive normalizing flow: For the contrastive normalizing flow (CF), we train solely with our new objective. We also run experiments using a finetuning training procedure (CF-FT), where we train a standard normalizing flow first and then use our new objective to finetune for two epochs.

**Baseline methods** We used the PyOD library (Zhao et al., 2019a) to compare our results to standard outlier detection techniques. We present results for PCA, KDE, and KNN conducted on the MoCo feature space, and show additional results for other methods and training directly on the image data (without feature extractor) in the appendix (see A.3). As a simple baseline, we use the mean squared error (MSE) to the mean of the feature space representations of the training set as an outlier score. This is equivalent to fitting a normal distribution to the feature space around the mean of the training data. This simple baseline already gives good results with the finetuned MoCo feature extractor. We extend the MSE to also employ the contrastive distribution by taking the difference of the MSE to the mean of the inlier set and the contrastive set as outlier score. This corresponds to the likelihood ratio of two normal distributions. We call this method MSE-ratio. As a third method (Flow), we train an unconditioned flow with the same architecture as our model on the MoCo feature representation and use the negative log-likelihood under the learned distribution as an outlier measure. As fourth baseline method (Flow-ratio), we train an additional flow (with the same architecture as our model) on the contrastive dataset and use the likelihood ratio of the learned inlier flow and the learned flow on the contrastive data using the hierarchies of distribution method of Schirrmeister et al. (2020). In contrast to their work, the normalizing flows are trained on the MoCo feature space and not on the images directly. We also employ the outlier exposure method of Hendrycks et al. (2019a) on our architecture by training a standard normalizing flow on the inlier data and finetune afterwards with their likelihood margin objective using the contrastive dataset. Finally, we compare our method to "CSI: Novelty Detection via Contrastive Learning on Distributionally Shifted Instances" (Tack et al., 2020). This method is the unsupervised state of the art method for one-class outlier detection on CIFAR-10. They train their model via contrastive learning, with transformed (shifted) instances of an image itself as additional negatives.

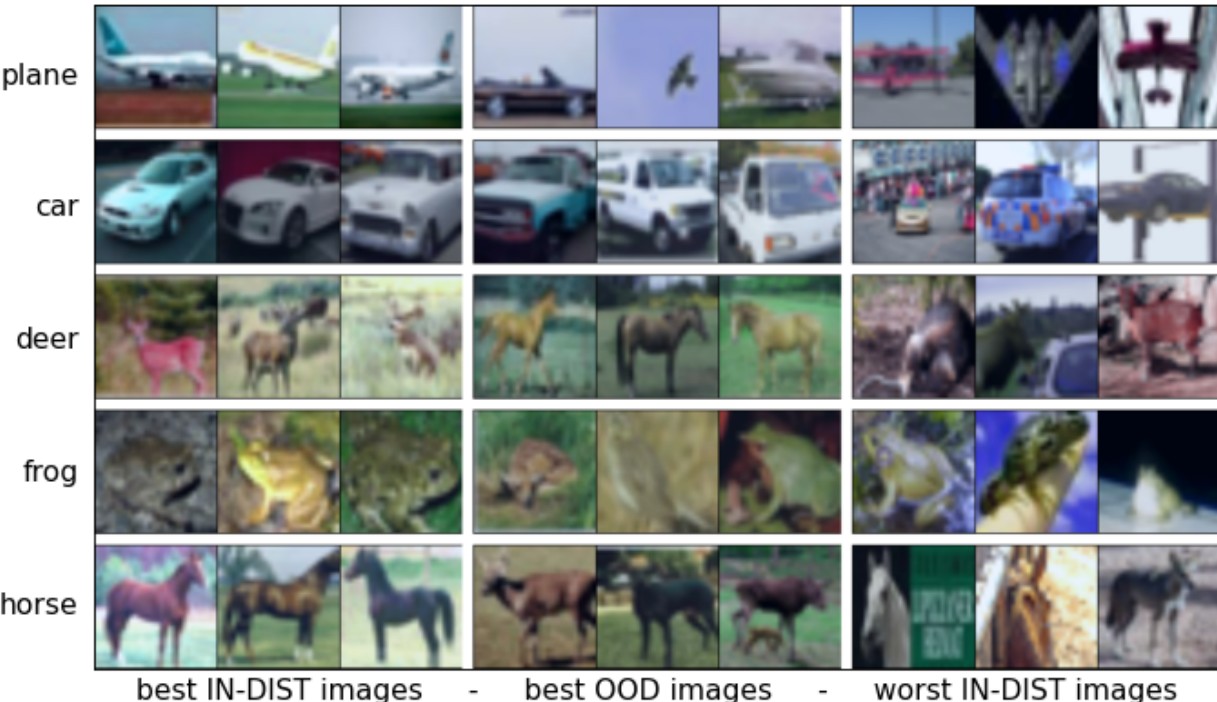

Figure 8: Visual examples from selected CIFAR-10 classes of the IN-DIST and OOD test sets. Every row shows a CIFAR-10 class as IN-DIST. From left to right: the three images with the lowest outlier score of the IN-DIST test set, the three images with the lowest outlier score of the OOD test set and the three images of the IN-DIST test set with the highest outlier score.

**Benchmark datasets** We conduct experiments using CIFAR-10, CIFAR-100 and CelebA as in-distribution. First, we run experiments on single classes of our datasets and compare at test time to the other classes not seen in training. We also conduct experiments in the dataset-vs-dataset setting, where we train on the whole CIFAR-10 dataset and compare to other datasets at test time.

### 5.3.1 One-vs-rest - results

We run experiments on CIFAR-10, CIFAR-100 (Krizhevsky, 2009) and celebA (Liu et al., 2015). We work in the one-vs-rest setting, where one class is used as inlier data, and all the other classes of the same dataset are used as anomalies at test time. For CIFAR-100, we show results for superclasses. For celebA, we divide the dataset into two classes by the given gender attribute and use one of the classes as inlier distribution and the other one as anomalies.

**CIFAR-10 and CIFAR-100 superclasses** We evaluate our method on the CIFAR-10 and CIFAR-100 datasets as inlier distribution with the IMAGENET dataset as contrastive distribution. Note that all CIFAR-10 and -100 classes are a subset of the IMAGENET dataset. The discussion in Section 4 holds also in practice: The broader contrastive IMAGENET distribution with overlap of the CIFAR inlier distributions does not negatively affect the performance, in contrary our method achieves state of the art results on almost all classes when evaluated under a One-vs-Rest setting. The quantative results of our model shown in figure 8 are sensible from a human perspective: The IN-DIST images with the highest outlier score show differ from typical IN-DIST images and some might be missclassified images of the CIFAR-10 dataset (e.g., the most anomalous frog and horse image in the figure). The OOD with the lowest outlier scores share semantic similarities with the IN-DIST class, e.g., cars and trucks or dogs and cat. We show quantitative results in table 2: One can see that our method is one of the best performing models in all settings, while the flow-ratio ablation method struggles in some settings and performs similarly to the simple flow approach. We think the

latter is a result of the training on the intermediate MoCo feature space. To further investigate our method, we show the confusion matrix for all classes in table 3: By looking at the failure cases, we can verify that the model focuses on semantic features: the two worst pairs are deer-horse and cat-dog, which are semantic similar classes. Truck and car have nearly perfect scores; only truck versus car fails (AUROC of about 90). This is also shown in figure 1, where we show the scores of all CIFAR-10 images for a model trained on the deer class as IN-DIST. We conduct the same experiments on the twenty CIFAR-100 superclasses and verify the CIFAR-10 results: The contrastive normalizing flow outperforms or is on par with the benchmarks and the baseline methods and reaches state of the art outlier detection performance. Our fine-tuned method CF-FT outperforms the benchmark methods. This advantage is statistically significant, we show a table of p-values in appendix A.6. The CIFAR-100 results for all superclasses and methods can be found in table 13 in the appendix.

Table 2: AUROC scores on CIFAR-10 classes for the One-Vs-Rest setting. PCA, KDE and KNN are taken from the PyOD library (Zhao et al., 2019a). GOAD from Bergman and Hoshen (2020), CSI from Tack et al. (2020), and Rot+T(rans) from Hendrycks et al. (2019b). MSE, MSE-ratio, and Flow are three ablation methods of our method contrastive normalizing flow (CF). Flow-ratio is the 'Hierarchies of Distributions'-method by Schirrmeister et al. (2020), but applied on the MoCo feature space. OE denotes the outlier exposure method by Hendrycks et al. (2019a). All methods within one standard deviation of the best AUROC score per class are highlighted. † denotes methods using an contrastive dataset.

| method | plane | car | bird | cat | deer |
|---|---|---|---|---|---|
| **KDE** | 94.4 | **99.1** | 90.4 | 90.4 | 93.3 |
| **PCA** | 94.2 | 98.9 | 90.6 | 90.4 | 93.2 |
| **KNN** | 96.0 | 98.7 | 92.3 | 90.4 | 93.6 |
| **GOAD** | 75.5 | 94.1 | 81.8 | 72.0 | 83.7 |
| **Rot+T** | 77.5 | 96.9 | 87.3 | 80.9 | 92.7 |
| **CSI** | 89.9 | **99.1** | 93.1 | 86.4 | **93.9** |
| **MSE** | 94.6 | **99.1** | 90.4 | 90.5 | 93.7 |
| **MSE-ratio** † | 92.8 | 98.5 | 89.8 | 89.7 | 91.8 |
| **Flow** | $96.1 \pm 0.2$ | $97.5 \pm 0.2$ | $92.6 \pm 0.2$ | $89.8 \pm 0.4$ | $93.3 \pm 0.3$ |
| **Flow-ratio** † | $95.9 \pm 0.2$ | $97.7 \pm 0.4$ | $93.5 \pm 0.5$ | $90.0 \pm 0.1$ | $93.2 \pm 0.5$ |
| **OE**† | $96.5 \pm 0.8$ | $\mathbf{99.2 \pm 0.1}$ | $\mathbf{92.9 \pm 1.7}$ | $92.6 \pm 0.6$ | $93.8 \pm 0.3$ |
| **CF (ours)** † | $96.9 \pm 0.3$ | $99.0 \pm 0.1$ | $\mathbf{94.6 \pm 0.1}$ | $92.8 \pm 0.4$ | $93.5 \pm 0.4$ |
| **CF-FT (ours)** † | $\mathbf{97.4 \pm 0.2}$ | $98.9 \pm 0.2$ | $\mathbf{94.9 \pm 0.5}$ | $\mathbf{93.5 \pm 0.6}$ | $\mathbf{94.8 \pm 0.6}$ |

| method | dog | frog | horse | ship | truck | mean |
|---|---|---|---|---|---|---|
| **KDE** | 92.1 | 96.2 | 94.7 | 98.5 | 98.2 | 94.7 |
| **PCA** | 93.0 | 96.6 | 95.0 | 98.6 | 98.3 | 94.9 |
| **KNN** | **95.7** | 98.0 | 95.8 | 98.5 | 97.3 | 95.6 |
| **GOAD** | 84.4 | 82.9 | 93.9 | 92.9 | 89.5 | 85.1 |
| **Rot+T** | 90.2 | 90.9 | 96.5 | 95.2 | 93.3 | 90.1 |
| **CSI** | 93.2 | 95.1 | **98.7** | 97.9 | 95.5 | 94.3 |
| **MSE** | 91.4 | 96.3 | 95.2 | **98.7** | 98.2 | 94.8 |
| **MSE-ratio** † | 92.5 | 95.4 | 94.3 | 98.3 | 97.6 | 94.1 |
| **Flow** | $\mathbf{95.7 \pm 0.4}$ | $\mathbf{98.0 \pm 0.4}$ | $94.7 \pm 0.4$ | $97.8 \pm 0.0$ | $96.6 \pm 0.1$ | $95.2 \pm 0.2$ |
| **Flow-ratio** † | $\mathbf{95.9 \pm 0.1}$ | $\mathbf{98.2 \pm 0.0}$ | $95.2 \pm 0.3$ | $97.5 \pm 0.4$ | $96.6 \pm 0.5$ | $95.4 \pm 0.2$ |
| **OE**† | $93.8 \pm 0.3$ | $97.6 \pm 0.4$ | $96.6 \pm 0.5$ | $98.4 \pm 0.1$ | $\mathbf{98.6 \pm 0.3}$ | $96.0 \pm 0.2$ |
| **CF (ours)** † | $\mathbf{96.1 \pm 0.4}$ | $\mathbf{98.2 \pm 0.0}$ | $96.3 \pm 0.2$ | $\mathbf{98.6 \pm 0.1}$ | $\mathbf{98.5 \pm 0.0}$ | $96.5 \pm 0.1$ |
| **CF-FT (ours)** † | $\mathbf{96.2 \pm 0.1}$ | $\mathbf{98.5 \pm 0.1}$ | $\mathbf{96.9 \pm 0.3}$ | $\mathbf{98.7 \pm 0.1}$ | $98.4 \pm 0.3$ | $\mathbf{96.8 \pm 0.1}$ |

Table 3: Confusion matrix on CIFAR-10 for the contrastive normalizing flow (CF). Every row shows results for a model trained on one Cifar-10 class as inlier distribution and evaluated against all other CIFAR-10 classes. Hard cases (AUROC<90) are printed bold.

|        | plane | car  | bird | cat  | deer | dog  | frog | horse | ship | truck | mean |
|--------|-------|------|------|------|------|------|------|-------|------|-------|------|
| plane  |       | 98.3 | 96.5 | 98.7 | 98.2 | 99.4 | 98.1 | 98.2  | **88.3** | 97.2 | 96.9 |
| car    | 99.6  |      | 99.9 | 99.8 | 99.9 | 99.9 | 99.8 | 99.8  | 99.2 | 93.0  | 99.0 |
| bird   | 94.9  | 99.9 |      | 94.6 | **83.5** | 97.3 | **88.8** | 93.3 | 99.3 | 99.8 | 94.6 |
| cat    | 97.9  | 99.7 | 93.9 |      | **88.7** | **72.3** | 90.9 | 92.0 | 99.1 | 99.7 | 92.8 |
| deer   | 97.7  | 99.4 | 92.3 | 93.8 |      | 96.1 | 93.2 | **70.5** | 98.8 | 99.7 | 93.5 |
| dog    | 99.5  | 99.6 | 98.3 | **82.9** | 95.8 |      | 98.6 | 91.2  | 99.4 | 99.8 | 96.1 |
| frog   | 98.8  | 99.7 | 96.8 | 95.9 | 95.9 | 98.8 |      | 99.4  | 99.1 | 99.9 | 98.2 |
| horse  | 98.6  | 99.8 | 97.1 | 96.4 | **82.3** | 94.4 | 99.0 |       | 99.4 | 99.8 | 96.3 |
| ship   | 93.8  | 98.3 | 99.6 | 99.4 | 99.5 | 99.6 | 99.5 | 99.5  |      | 98.4 | 98.6 |
| truck  | 98.4  | 91.2 | 99.8 | 99.6 | 99.7 | 99.8 | 99.7 | 99.5  | 98.7 |      | 98.5 |

**CelebA** In the following experiment we want to measure the performance for a contrastive distribution dissimilar to the in-distribution data on real world data: We evaluate our model on the CelebA dataset. We use the gender attribute given in the dataset to divide the dataset in two classes, and use each individually as inlier distribution and the other class as test-time outlier distribution. Even though our contrastive distribution for training the contrastive normalizing flow and the flow-ratio method is the IMAGENET dataset, which does not contain close-up pictures of human faces, our method beats the baselines. However, the lacking overlap of the contrastive distribution with the test-time outlier distribution explains why the contrastive approaches such as the flow-ratio method and our contrastive flow do not outperform the other baselines by a clear margin and the overall performance leaves room for improvement. The fine-tuned method CF-FT does not show the same performance improvement as in the CIFAR-10 and CIFAR-100 experiments. We report the results in table 4.

Table 4: AUROC scores of models trained on the celebA dataset split into the provided gender attribute and evaluated against the other class using contrastive normalizing, CSI, and our ablation methods. MCD and KNN are taken from the PyOD library (Zhao et al., 2019a). All models within one standard deviation of the best performing model are highlighted. We neglect the small standard deviation of the deterministic methods introduced by the MoCo preprocessing. Please note that the model only sees images of the in-distribution class and the unlabeled contrastive IMAGENET dataset at training time.

| **IN-DIST** | MCD | KNN | CSI | Flow | Ratio | OE | **CF (our)** | **CF-FT (our)** |
|-------------|------|------|------|------|------|------|------|------|
| female      | 69.7 | 74.1 | 68.3 | $75.8 \pm 2.1$ | $76.7 \pm 1.8$ | $80.3 \pm 1.2$ | $\mathbf{83.8 \pm 3.2}$ | $77.3 \pm 1.5$ |
| male        | 76.6 | 82.3 | 79.1 | $\mathbf{86.7 \pm 0.8}$ | $\mathbf{86.4 \pm 2.3}$ | $\mathbf{88.2 \pm 1.4}$ | $87.1 \pm 2.6$ | $\mathbf{86.9 \pm 1.8}$ |
| average     | 73.2 | 78.2 | 73.7 | $81.3 \pm 1.1$ | $81.5 \pm 1.5$ | $\mathbf{84.3 \pm 0.9}$ | $\mathbf{85.3 \pm 2.1}$ | $82.1 \pm 1.2$ |

### 5.3.2 Dataset-vs-dataset

Next we investigate the setting of multi-modal in-distribution data and distant OOD samples. For the dataset-vs-dataset setting, the complete unlabeled dataset is used as the inlier distribution at training time. We train a contrastive normalizing flow on CIFAR-10 and use CIFAR-100, celebA and SVHN as outlier distributions at test time. We report the AUROC scores in table 5. This setting is well studied, e.g., by Nalisnick et al. (2019a).

Table 5: AUROC scores in the dataset-vs-dataset setting. We train on CIFAR-10 as IN-DIST and IMA-GENET as contrastive distribution. Additional results are reported in table 10. The standard deviations are taken over three runs. All models within one standard deviation of the best performing model are highlighted.

| method | CIFAR10 vs CIFAR100 | CIFAR10 vs SVHN | CIFAR10 vs celebA | mean |
|---|---|---|---|---|
| MSE | $70.0 \pm 0.0$ | $20.0 \pm 0.0$ | $92.2 \pm 0.0$ | $60.7 \pm 0.0$ |
| MSE-Ratio† | $74.8 \pm 0.0$ | $42.0 \pm 0.0$ | $91.5 \pm 0.0$ | $69.4 \pm 0.0$ |
| Flow | $82.9 \pm 0.6$ | $65.4 \pm 3.3$ | $\mathbf{100 \pm 0.1}$ | $82.8 \pm 1.1$ |
| Flow-ratio † | $\mathbf{84.7 \pm 0.3}$ | $68.3 \pm 6.6$ | $\mathbf{100 \pm 0.0}$ | $83.3 \pm 2.2$ |
| OE† | $81.3 \pm 0.9$ | $65.0 \pm 2.3$ | $\mathbf{99.9 \pm 0.1}$ | $82.8 \pm 0.8$ |
| CF (ours) † | $\mathbf{84.4 \pm 0.3}$ | $\mathbf{90.3 \pm 1.5}$ | $\mathbf{99.8 \pm 0.3}$ | $\mathbf{91.5 \pm 0.5}$ |
| CF-FT (ours) † | $\mathbf{84.7 \pm 0.2}$ | $84.7 \pm 1.4$ | $\mathbf{99.7 \pm 0.4}$ | $89.6 \pm 0.5$ |

### 5.3.3  Discussion of benchmark experiments

We investigate the performance of our difference method in benchmark experiments on CIFAR-10, CIFAR-100 and CelebA. We find that our model is for almost all settings the best performing or within one standard deviation of the best performing method. We report a confusion matrix in table 3, where we see that most difficult pairs of inlier and outlier data are semantically similar classes (e.g., horse-deer or cat-dog). We show in the appendix in section A.10 in-distribution images with high outlier score and out-of-distribution images with low outlier score. We argue that these images are also for humans hard to decide and some of them may be missclassified in the original CIFAR-10 dataset. We see for the CIFAR-10 and CIFAR-100 single-class experiments a significant performance gain for our fine-tuned method CF-FT, for the larger and more diverse in-distribution datasets (CelebA and dataset setting) the contrastive flow fully trained with our new objective shows the best performance.

## 6  Conclusion

We propose a novel method to train density estimators. The method relies on the use of a contrastive dataset for training: The training objective maximizes the log-likelihood of in-distribution data while minimizing the log-likelihood on the contrastive dataset. We show that using our training objective, the density estimator learns the normalized positive difference of the in-distribution and the contrastive distribution. We use the new objective to train normalizing flows and show its application to outlier detection. To increase the performance of the method and reduce the computational cost, we employ a pretrained feature extractor on which the contrastive normalizing flow operates. We study the influence of the constrastive distribution in several experiments and compare to the ratio method Ren et al. (2019) and the standard normalizing flow without contrastive dataset. We find that our model is robust for sensible choices of the contrastive distribution and alleviates theoretical problems of the ratio method also in the experiments. Furthermore, we show that our method outperforms state-of-the-art methods from the literature on several benchmark datasets. In this study, we utilized the contrastive normalizing field solely for outlier detection. However, contrastive normalizing flows have diverse potential applications such as dataset reduction and learning the difference distribution between two datasets. To comprehend the high-dimensional difference distributions, more research is necessary.

### Acknowledgments

We want to thank all our TMLR reviewers, in particular Reviewer wnbF from the first submission for the comments and questions which helped to significantly strengthen the paper. Furthermore we want to thank Felix Draxler, Dan Zhang and Philipp Geiger for proofreading the manuscript and their helpful suggestions. We would like to acknowledge the TMLR editorial team for their efficient and professional handling of the submission process.

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

# A  Appendix

## A.1  Training Pseudocode

To clarify how the model is trained, we report the training process in pseudocode in algorithm 1.

---
**Algorithm 1** Training process

---
With MoCo feature extractor, density estimator $T_\theta$ and $\text{nll}_\theta(\boldsymbol{x}) = -\log \text{probability}_{T_\theta}(\boldsymbol{x})$
**for all** epochs **do**
    **for all** batches $\boldsymbol{x} \sim$ inlier data and $\boldsymbol{y} \sim$ contrastive data **do**
        **for all** $\boldsymbol{x}_i \sim \boldsymbol{x}$ and $\boldsymbol{y}_j \sim \boldsymbol{y}$ **do**
            $\hat{\boldsymbol{x}}_i = \text{MoCo}(\boldsymbol{x}_i), \hat{\boldsymbol{y}}_j = \text{MoCo}(\boldsymbol{y}_j)$
            $\text{lossPos}_i = \text{nll}_\theta(\hat{\boldsymbol{x}}_i)$
            $\text{lossNeg}_j = \text{nll}_\theta(\hat{\boldsymbol{y}}_j)$
            $\text{lossNeg}_i = \text{clamp}(\text{lossNeg}_i, \text{None}, \text{tsh})$
        **end for**
        $\text{loss} = \frac{1}{N}\sum_i^N \text{lossPos}_i - \frac{1}{M}\sum_j^M \text{lossNeg}_j$
        $\text{GradientStep}(\theta, \text{loss})$
    **end for**
**end for**

---

## A.2  Implementation details

### A.2.1  Contrastive Loss Clamping

For our loss, we apply a small deviation from the theory: The second term in equation 6 results in the theoretical optimum $p_\theta(\boldsymbol{x}) = 0 \ \forall \boldsymbol{x} \in \text{supp}(q > p)$, but the loss diverges because $\lim_{x \to 0} \log x$ is unbounded. To handle this mismatch, we clamp $\log p(\boldsymbol{x})$ at a threshold $\epsilon$ as a lower bound on the likelihood. Therefore the objective leads to $\log p(\boldsymbol{x}) < \epsilon \ \forall \boldsymbol{x} \in \text{supp}(q > p)$, which we find to be sufficient for successful outlier detection. For our experiments, we found $\epsilon = 0$ sufficient for a good outlier detection performance. For unknown datasets, we recommend to train a standard normalizing flow first and using the distribution of the in-distribution likelihoods to determine a constant clamping parameter, e.g. the 10% quantile of the in-distribution log density plus a fixed offset (e.g., $\ln(10) = 2.3$) . We investigate the influence of the threshold $\epsilon$ in the appendix in A.4 and find that the outlier detection performance is constant under reasonable changes of $\epsilon$.

### A.2.2  MoCo Finetuning

For datasets with image sizes significantly smaller than the images used in the MoCo implementation with 224 by 224 pixels, the features are dominated by upsampling artefacts and are highly correlated between different images. To reduce this problem without changing the setup, we finetuned a MoCo trained on 244*244 images on smaller images by shrinking IMAGENET images to 32 by 32 pixels and then using upsamling to 224 by 224 pixels again. We discuss this in appendix A.4.

### A.2.3  Hyperparameter settings

As our contrastive normalizing flow, a standard normalizing flow and the ratio method use the same architecture and similar training procedures, we decided to use a standard architecture similiar to the architecture proposed in `https://vislearn.github.io/FrEIA/_build/html/tutorial/examples/fully_connected.html`. We used the AUROC score between in-distribution validation set and imagenet validation set (contrastive) as proxy for the outlier performance and tuned our method and all our ablation methods. For our contrastive normalizing flow, the reported hyperparameter worked best over all real world data experiments. All tested models showed to be robust on small hyperparameter changes. We also used this proxy AUROC to implement early stopping and prevent overfitting.

Table 6: Hyperparameter settings for all contrastive normalizing flow experiments. CF stands for our contrastive normalizing flow model.

| Hyperparameter | |
|---|---|
| Batchsize | 256 |
| Optimizer | Adam |
| Learning rate | $10^{-3}$ |
| CF implementation | Freia (Ardizzone et al., 2018-2023) |
| CF architecture | 8 Freia AllInOneBlocks |
| CF subnetwork | MLP with 512 hidden layers and ReLU act. |
| CF affine clamping | 3. |
| CF dimensions | 128 (MoCo output) |
| Contrastive loss clamping | 0. |
| MoCo implementation | He et al. (2020) |
| MoCo finetuning | Added 32*32 downsampling to preprocessing |

Table 7: AUROC for different methods of the PyOD (Zhao et al., 2019a) library on the CIFAR-10 one-vs-rest setting with standard deviation over three runs. The first column shows the results on the image data, the second on unnormalized MoCo features and the last on normalized MoCo features. Using normalized MoCo features gives the best results for most methods. The three best performing methods KDE, PCA and KNN were reported in table 2. The small standard deviations for the deterministic models (COPOD, LOF, MCD, ABOD, KDE, PCA and KNN) are introduced by the MoCo preprocessing and upsampling. They are reduced by the normalization, such that the rounded values are 0.0 for the last four methods.

| Method | No Feature extractor | FE + No Normalization | FE + Normalization |
|---|---|---|---|
| COPOD | $54.8 \pm 0.0$ | $63.0 \pm 0.1$ | $86.3 \pm 0.1$ |
| INNE | $57.6 \pm 0.7$ | $77.9 \pm 0.7$ | $88.6 \pm 0.6$ |
| LOF | $57.5 \pm 0.0$ | $94.2 \pm 0.1$ | $89.9 \pm 0.1$ |
| MCD | — | $92.6 \pm 0.0$ | $91.6 \pm 0.1$ |
| iForest | $59.8 \pm 0.2$ | $79.5 \pm 0.6$ | $92.2 \pm 0.2$ |
| Sampling | $58.8 \pm 1.2$ | $82.0 \pm 1.6$ | $93.5 \pm 0.3$ |
| ABOD | $58.7 \pm 0.0$ | $92.9 \pm 0.1$ | $94.0 \pm 0.0$ |
| KDE | $54.3 \pm 0.0$ | $77.3 \pm 0.1$ | $94.7 \pm 0.0$ |
| PCA | $58.7 \pm 0.0$ | $79.4 \pm 0.1$ | $94.9 \pm 0.0$ |
| KNN | $59.1 \pm 0.0$ | $91.5 \pm 0.1$ | $95.6 \pm 0.0$ |
| CF (ours) | — | $95.8 \pm 0.2$ | $96.5 \pm 0.1$ |

### A.3 PyOD Experiments

We run additional experiments for methods from the PyOD library (Zhao et al., 2019a) without feature extractor and with feature extractor with and without normalization of the features (the results in the main paper are with feature extractor and normalization). The results are reported in table 7, 8, 9, and 10. The small standard deviations for the deterministic models (COPOD, LOF, MCD, ABOD, KDE, PCA and KNN) are introduced by the MoCo preprocessing and upsampling. The experiments show that using the MoCo feature space significantly improves the outlier detection performance for all models. Normalizing the features improves the score for most models.

### A.4 Ablation studies

**Tuning the Threshold** $\epsilon$ To show the dependence of our model on the clamping threshold $\epsilon$, we run two ablation studies: First, we learn one-dimensional contrastive normalizing flows on the experimental setup of our toy experiment in section 4. As we know the ground truth difference, we choose different thresholds

Table 8: AUROC for different methods of the PyOD (Zhao et al., 2019a) library on the CIFAR-100 one-vs-rest setting with standard deviation over three runs. The first column shows the results on the image data, the second on unnormalized MoCo features and the last on normalized MoCo features. Using normalized MoCo features gives the best results most methods. The three best performing methods KDE, PCA and KNN were reported in table 2. The small standard deviations for the deterministic models (COPOD, LOF, MCD, ABOD, KDE, PCA and KNN) are introduced by the MoCo preprocessing and upsampling. They are reduced by the normalization, such that the rounded values are 0.0 for the last three methods.

| Method | No Feature Extractor | FE + No Normalization | FE + Normalization |
|---|---|---|---|
| COPOD | $53.0 \pm 0.0$ | $54.0 \pm 0.1$ | $73.4 \pm 0.2$ |
| INNE | $57.2 \pm 0.7$ | $69.9 \pm 0.4$ | $83.3 \pm 1.0$ |
| LOF | $59.9 \pm 0.0$ | $90.1 \pm 0.1$ | $87.2 \pm 0.1$ |
| iForest | $59.5 \pm 0.3$ | $71.2 \pm 0.4$ | $89.3 \pm 0.2$ |
| Sampling | $58.4 \pm 0.4$ | $77.1 \pm 0.7$ | $91.2 \pm 0.5$ |
| MCD | — | $91.5 \pm 0.0$ | $91.5 \pm 0.2$ |
| ABOD | $60.2 \pm 0.0$ | $89.9 \pm 0.1$ | $92.8 \pm 0.1$ |
| KDE | $54.0 \pm 0.0$ | $67.9 \pm 0.0$ | $93.0 \pm 0.0$ |
| PCA | $57.0 \pm 0.0$ | $70.5 \pm 0.1$ | $93.2 \pm 0.0$ |
| KNN | $60.0 \pm 0.0$ | $86.4 \pm 0.0$ | $94.8 \pm 0.0$ |
| CF (ours) | — | $93.8 \pm 0.2$ | $95.1 \pm 0.1$ |

Table 9: AUROC for different methods of the PyOD (Zhao et al., 2019a) library on the celebA class-vs-class setting with standard deviation over three runs. The first column shows the results on the unnormalized MoCo features and the second on normalized MoCo features.

| | FE + No Normalization | | FE + Normalization | |
|---|---|---|---|---|
| Method | male | female | male | female |
| LOF | $67.6 \pm 0.5$ | $61.6 \pm 0.1$ | $53.7 \pm 0.3$ | $55.1 \pm 0.1$ |
| ABOD | $82.1 \pm 0.3$ | $63.7 \pm 0.1$ | $68.9 \pm 0.4$ | $76.4 \pm 0.1$ |
| MCD | $79.0 \pm 0.2$ | $67.6 \pm 0.0$ | $69.7 \pm 0.0$ | $76.6 \pm 0.2$ |
| LODA | $74.8 \pm 2.9$ | $57.9 \pm 5.8$ | $81.2 \pm 1.5$ | $77.9 \pm 4.0$ |
| KNN | $77.2 \pm 0.1$ | $70.1 \pm 0.2$ | $74.0 \pm 0.1$ | $82.3 \pm 0.0$ |

Table 10: AUROC for different methods of the PyOD (Zhao et al., 2019a) library on the CIFAR-10 vs. dataset setting with standard deviation over three runs. The first column shows the results on the unnormalized MoCo features and the second on normalized MoCo features.

| | FE + No Normalization | | | FE + Normalization | | |
|---|---|---|---|---|---|---|
| Method | CIFAR-100 | SVHN | celebA | CIFAR-100 | SVHN | celebA |
| KNN | $76.4 \pm 0.1$ | $29.5 \pm 0.6$ | $98.91 \pm 0.0$ | $83.8 \pm 0.1$ | $89.6 \pm 0.3$ | $99.9 \pm 0.0$ |
| LOF | $81.4 \pm 0.0$ | $44.2 \pm 2.6$ | $99.8 \pm 0.2$ | $71.3 \pm 0.0$ | $33.7 \pm 1.2$ | $95.3 \pm 0.2$ |
| iForest | $52.0 \pm 1.0$ | $8.5 \pm 0.0$ | $70.4 \pm 0.4$ | $68.6 \pm 0.7$ | $56.0 \pm 10.4$ | $65.9 \pm 2.5$ |
| LODA | $51.2 \pm 2.5$ | $12.2 \pm 2.8$ | $61.3 \pm 4.7$ | $60.5 \pm 2.3$ | $51.8 \pm 3.2$ | $81.2 \pm 8.7$ |
| MCD | $71.8 \pm 2.5$ | $66.0 \pm 1.9$ | $97.1 \pm 2.2$ | $64.5 \pm 7.7$ | $74.5 \pm 12.0$ | $83.4 \pm 17.0$ |

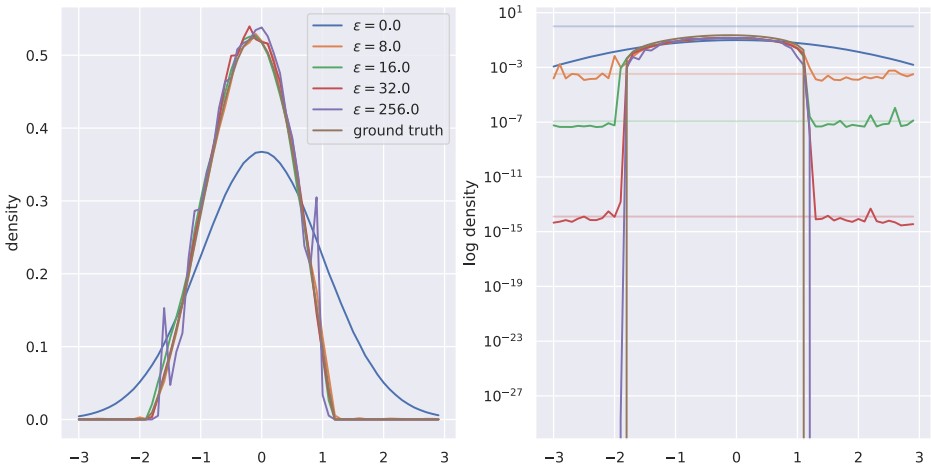

Figure 9: Density plots for models trained on samples $x \sim p = N(0, 1)$ as in-distribution and $y \sim q = N(1, 2)$ as contrastive dataset for different thresholds $\epsilon$. On the right, the densities and thresholds are plotted in log space. Because $\epsilon = 0.0$ is larger than the in-distribution log density, the contrastive dataset is ignored while training and the in-distribution density is learned. For large $\epsilon$ the approximation of the in-distribution suffers.

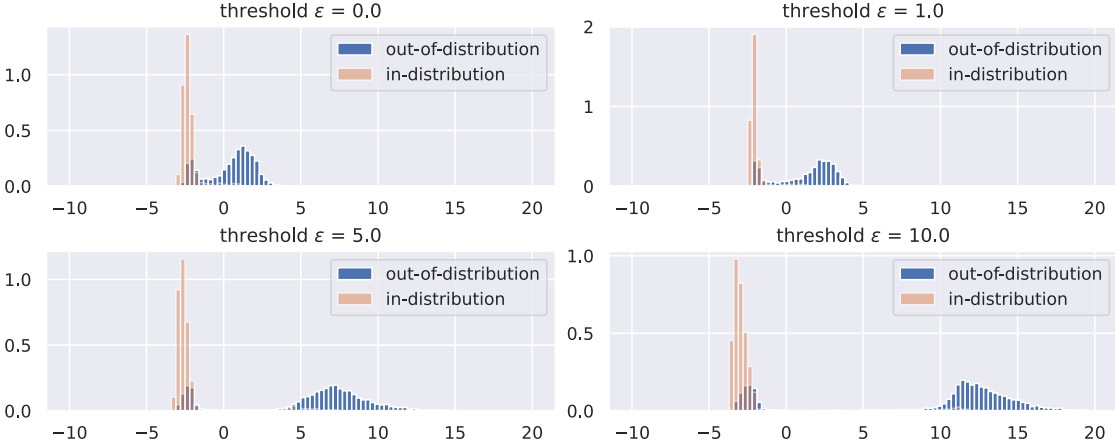

Figure 10: Histograms for in-distribution and out-of-distribution outlier scores for contrastive normalizing flows trained on the CIFAR-10 cat class as in-distribution, with different thresholds $\epsilon$. The AUROC score is constant over all models ($92.8 \pm 0.4$)

ranging from values larger than the ground truth log like-likelihood ($\epsilon = 0.0$) to values orders of magnitude smaller ($\epsilon = 100.0$). We show the learned densities of our models in figure 9. As one can see, the models approximate the ground truth density. If the thresholds gets too small, no contrastive data is used in the objective and the in-distribution density is learned. If the threshold is too large, the approximation for high likelihoods worsens as the model needs to model the tails of the distribution.

We also run an ablation study on the cat class of the CIFAR-10 dataset. We plot in figure 10 histograms for the in-distribution and out-of-distribution (all other CIFAR-10-classes) outlier scores for models trained with different thresholds $\epsilon$. As simple ood-cases are pushed above the threshold, but hard ood-cases overlap with the in-distribution scores for all thresholds, the performance stays constant for all four models.

| IN-DIST | Flow (inceptionV3) | Ratio (inceptionV3) | CF (ours, inceptionV3) | CF (ours, MoCo) |
|---------|--------------------|--------------------|-----------------------|-----------------|
| male | $74.4 \pm 1.1$ | $75.1 \pm 2.7$ | $78.2 \pm 3.8$ | $83.8 \pm 3.2$ |
| female | $82.0 \pm 2.4$ | $82.0 \pm 2.8$ | $83.1 \pm 2.4$ | $87.1 \pm 2.6$ |
| average | $78.7 \pm 1.3$ | $78.6 \pm 2.0$ | $80.7 \pm 2.2$ | $85.3 \pm 2.1$ |

Table 11: Performance of our model and ablation methods on the (projected) inceptionV3 feature space.

Table 12: Ablation for normalization of the MoCo features. Normalization the features improves clustering, as one can see by the improvement on our MSE baseline.

|  | Without Norm. | | With Norm. | |
|---------|------|------|------|------|
| dataset | MSE | CF | MSE | CF |
| CIFAR-10 | 77.1 | 95.8 | 94.3 | **96.5** |
| CIFAR-100 | 67.6 | 93.8 | 92.2 | **95.1** |

**No normalization of the features**   Because of the cosine similarity in the contrastive learning objective, the contrastive loss is invariant under changes of the norm of the feature vectors. For the normalizing flow, the norm of the input vector has an important influence on the likelihood; therefore we decided to normalize all feature vectors to account for this mismatch. While this changes the topology of the feature space drastically, the normalization improves the performance of our method. The results can be found in table 12. We show the performance for the contrastive normalizing flow and the MSE ablation method because it directly shows the clustering of the features.

**MoCo finetuning**   The original MoCo is trained on IMAGENET, with image input sizes of 244*244 pixels. When using the feature extractor on smaller images like the CIFAR-10 or CIFAR-100 dataset, the MoCo features are dominated by upsampling artifacts: The feature dimensions of upsampled images have a strong correlation and are mostly independent of the semantic content. To improve the performance of the model, we finetuned the feature extractor with additional MoCo training on first down- and then upsampled IMAGENET data. The resulting features are less correlated and the performance of models trained on the resulting feature spaces is significantly improved. Examples for both feature spaces are shown in figure 11.

**Use of a different feature extractor**   We use the inceptionV3 feature extractor, used for the calculation of FID-scores, to investigate the dependence of our model on the MoCo feature extractor used in all our experiments. As inceptionV3 is trained as a classifier, we use the penultimate layer for feature extraction. We use the celebA dataset for comparison, as the inceptionV3 model only works for inputs of size 299 by 299 pixels or larger. To use the same architecture as for our MoCo experiments, we project the 2048 dimensional inceptionV3 feature space on 128 dimensions by summing over fixed groups of 16 features. We show the results in table 11 and compare to the performance on the MoCo feature space. The performance drops for all methods by about 4 AUROC compared to the experiments on the MoCo feature space. This experiment demonstrates that the advantage of our method is not feature space dependent and that the MoCo feature extractor is a good choice for all methods.

## A.5   CIFAR-100 results and standard deviations

We report in table 13 the AUROCS for all CIFAR-100 clases for all methods and in table 14 the standard deviations for our methods.

## A.6   Significance tests

Following the procedure in Demšar (2006), we test the significance of our method against the benchmark methods with the Wilcoxon signed ranks test (Wilcoxon, 1945). We use the means over three runs to compare the different methods. The p-values for the null-hypothesis "our model does not outperform the benchmark method" are shown in figure 12 (CIFAR-10) and figure 13 (CIFAR-100). The fine-tuned methods outperforms

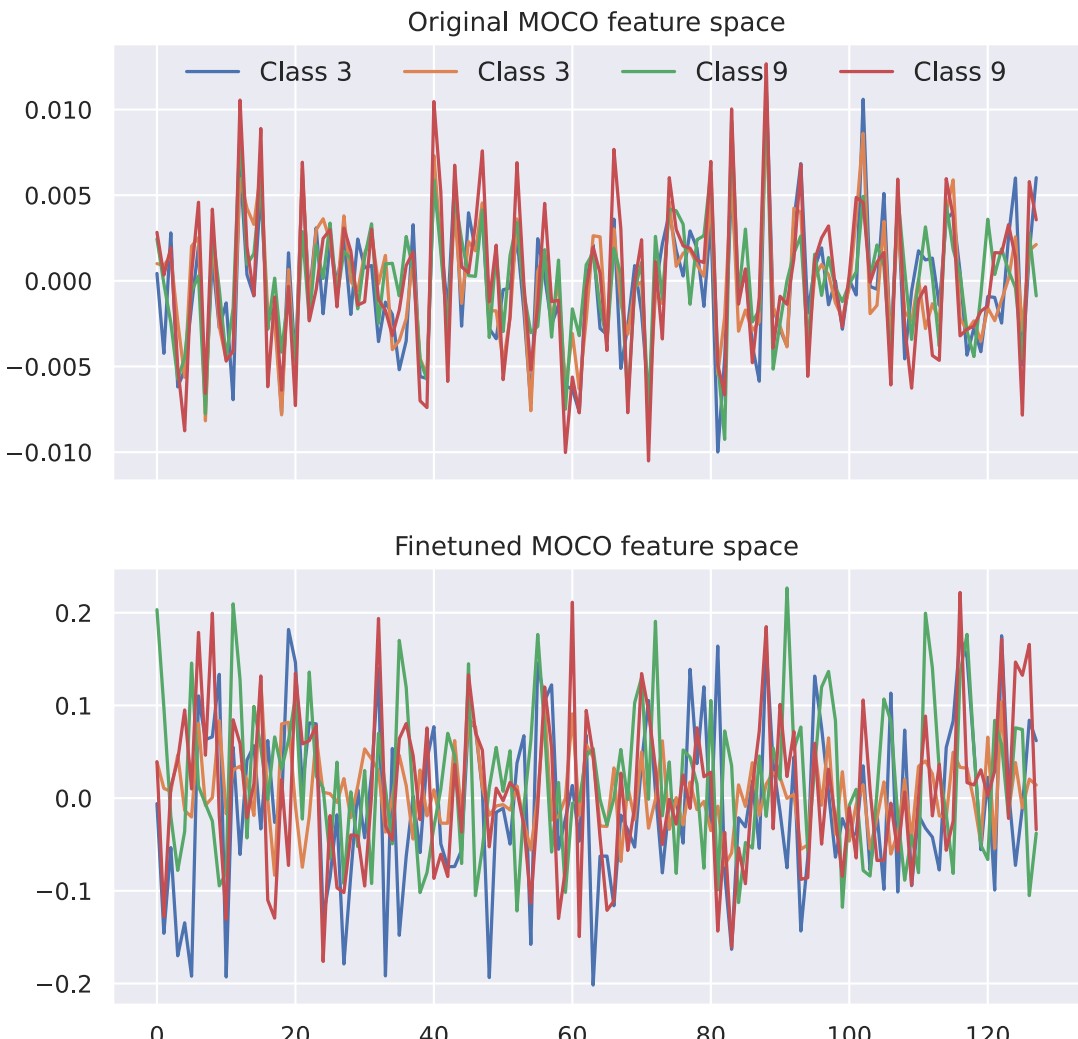

Figure 11: Qualitative comparison of the original and finetuned MoCo feature space. In the original MoCo feature space, the dimensions have a strong correlation for images of different classes, while in the finetuned MoCo feature space the feature dimensions are not correlated across different classes.

Table 13: AUROC scores on CIFAR-100 superclasses for the One-Vs-Rest setting. Our method contrastive normalizing flow (CF) hast the best outlier detection performance over all classes. Best AUROC scores per class are bold. † denotes methods using a contrastive dataset.

| method | 0 | 1 | 2 | 3 | 4 | 5 | 6 | 7 | 8 | 9 |
|---|---|---|---|---|---|---|---|---|---|---|
| **KDE** | 90.7 | 91.0 | 96.4 | 95.2 | 95.7 | 90.9 | 96.4 | 92.7 | 92.6 | 96.5 |
| **PCA** | 91.2 | 91.0 | 96.6 | 95.9 | 95.6 | 90.9 | 96.1 | 93.2 | 92.9 | 96.7 |
| **KNN** | 91.6 | 92.7 | 96.3 | **97.1** | 97.2 | **95.4** | 97.2 | 94.5 | 95.6 | 96.5 |
| **CSI** | 86.3 | 84.8 | 88.9 | 85.7 | 93.7 | 81.9 | 91.8 | 83.9 | 91.6 | 95.0 |
| **MSE** | 90.0 | 90.5 | 96.4 | 94.5 | 94.9 | 89.3 | 96.0 | 92.0 | 91.7 | 96.0 |
| **MSE-ratio †** | 90.5 | 89.5 | 95.8 | 95.3 | 94.7 | 90.6 | 95.9 | 92.6 | 93.4 | 96.5 |
| **Flow** | 91.7 | 92.7 | 91.7 | 92.7 | 91.7 | 92.7 | 91.7 | 92.7 | 91.7 | 92.7 |
| **Flow-ratio †** | 93.1 | 92.9 | 96.0 | 96.5 | 97.0 | 94.5 | 97.2 | 94.5 | 95.1 | 95.9 |
| **OE †** | 93.3 | 94.2 | 94.8 | 96.8 | 97.1 | 94.4 | 95.5 | 94.9 | 95.3 | 97.2 |
| **CF (ours) †** | 92.6 | 94.1 | 97.3 | 95.9 | 96.8 | 95.0 | 97.3 | 94.3 | 95.0 | 97.1 |
| **CF-FT (ours) †** | **93.8** | **94.9** | **97.9** | **97.1** | **97.4** | **95.4** | **97.6** | **95.3** | **95.8** | **97.8** |

| method | 10 | 11 | 12 | 13 | 14 | 15 | 16 | 17 | 18 | 19 | mean |
|---|---|---|---|---|---|---|---|---|---|---|---|
| **KDE** | 96.1 | 92.0 | 88.7 | 88.3 | 96.7 | 88.8 | 85.9 | 97.8 | 96.4 | 91.5 | 93.0 |
| **PCA** | 96.3 | 92.0 | 89.5 | 88.0 | 96.7 | 88.6 | 86.9 | 97.8 | 96.0 | 91.3 | 93.2 |
| **KNN** | **97.4** | **94.6** | 92.3 | 90.6 | 98.5 | 88.8 | 90.7 | 97.8 | **97.3** | 94.0 | 94.8 |
| **CSI** | 94.0 | 90.1 | 90.3 | 81.5 | 94.4 | 85.6 | 83.0 | 97.5 | 95.9 | 95.1 | 89.6 |
| **MSE** | 95.8 | 90.7 | 88.0 | 87.0 | 95.5 | 87.0 | 85.5 | 97.6 | 95.2 | 89.4 | 92.2 |
| **MSE-ratio†** | 95.3 | 92.7 | 88.8 | 87.9 | 96.7 | 88.7 | 85.7 | 96.1 | 95.9 | 90.2 | 92.3 |
| **Flow** | 96.3 | 93.2 | 90.1 | 91.9 | 97.9 | 88.8 | 90.3 | 97.4 | 96.7 | 94.7 | 93.0 |
| **Flow-ratio†** | 96.1 | 93.6 | 89.7 | 91.7 | 92.6 | 89.1 | 91.0 | 98.0 | 96.8 | 95.0 | 94.6 |
| **OE †** | **97.4** | 94.2 | 92.0 | **92.0** | 98.4 | 91.5 | 91.3 | 98.4 | 97.0 | 95.2 | 95.0 |
| **CF (ours)†** | 97.0 | 94.5 | 91.4 | 91.7 | 97.9 | 90.9 | 91.4 | 98.4 | 96.1 | 95.6 | 95.0 |
| **CF-FT (ours)†** | 97.1 | 94.4 | **92.4** | 91.7 | **98.6** | **92.0** | **92.2** | **98.5** | 97.2 | **96.5** | **95.7** |

Table 14: Standard deviations over three runs for our method Contrastive normalizing flow (CF) and the baseline methods 'Flow' and 'Flow-ratio' for all CIFAR-100 classes.

| class | Flow | Flow-ratio | CF (ours) | CF-FT (ours) |
|---|---|---|---|---|
| 0 | $91.7 \pm 0.3$ | $93.1 \pm 0.6$ | $92.6 \pm 0.4$ | $93.8 \pm 0.0$ |
| 1 | $92.7 \pm 0.9$ | $92.9 \pm 0.9$ | $94.1 \pm 0.2$ | $94.9 \pm 0.4$ |
| 2 | $91.7 \pm 0.3$ | $96.0 \pm 0.1$ | $97.3 \pm 0.2$ | $97.9 \pm 0.0$ |
| 3 | $92.7 \pm 0.4$ | $96.5 \pm 1.0$ | $95.9 \pm 0.5$ | $97.1 \pm 0.2$ |
| 4 | $91.7 \pm 0.5$ | $97.0 \pm 0.3$ | $96.8 \pm 0.2$ | $97.4 \pm 0.4$ |
| 5 | $92.7 \pm 0.4$ | $94.5 \pm 0.3$ | $95.0 \pm 0.1$ | $95.4 \pm 0.2$ |
| 6 | $91.7 \pm 0.2$ | $97.2 \pm 0.2$ | $97.3 \pm 0.6$ | $97.6 \pm 0.4$ |
| 7 | $92.7 \pm 0.2$ | $94.5 \pm 0.5$ | $94.3 \pm 0.2$ | $95.3 \pm 0.4$ |
| 8 | $91.7 \pm 0.3$ | $95.1 \pm 0.8$ | $95.0 \pm 0.1$ | $95.8 \pm 0.1$ |
| 9 | $92.7 \pm 0.0$ | $95.9 \pm 0.3$ | $97.1 \pm 0.6$ | $97.8 \pm 0.3$ |
| 10 | $96.3 \pm 0.6$ | $96.1 \pm 0.6$ | $97.0 \pm 0.6$ | $97.1 \pm 0.1$ |
| 11 | $93.2 \pm 0.0$ | $93.6 \pm 0.7$ | $94.5 \pm 0.2$ | $94.4 \pm 0.4$ |
| 12 | $90.1 \pm 1.4$ | $89.7 \pm 0.5$ | $91.4 \pm 0.6$ | $92.4 \pm 0.2$ |
| 13 | $91.9 \pm 0.4$ | $91.7 \pm 0.5$ | $91.7 \pm 0.7$ | $91.7 \pm 0.3$ |
| 14 | $97.9 \pm 0.5$ | $92.6 \pm 0.5$ | $97.9 \pm 0.7$ | $98.6 \pm 0.4$ |
| 15 | $88.8 \pm 0.7$ | $89.1 \pm 0.2$ | $90.9 \pm 0.5$ | $92.0 \pm 0.3$ |
| 16 | $90.3 \pm 0.9$ | $91.0 \pm 0.5$ | $91.4 \pm 0.5$ | $92.2 \pm 0.3$ |
| 17 | $97.4 \pm 0.3$ | $98.0 \pm 0.2$ | $98.4 \pm 0.1$ | $98.5 \pm 0.2$ |
| 18 | $96.7 \pm 0.3$ | $96.8 \pm 0.5$ | $96.1 \pm 0.6$ | $97.2 \pm 0.6$ |
| 19 | $94.7 \pm 0.4$ | $95.0 \pm 0.3$ | $95.6 \pm 0.5$ | $96.5 \pm 0.1$ |
| mean | $93.0 \pm 0.2$ | $94.6 \pm 0.2$ | $95.0 \pm 0.1$ | $95.7 \pm 0.1$ |

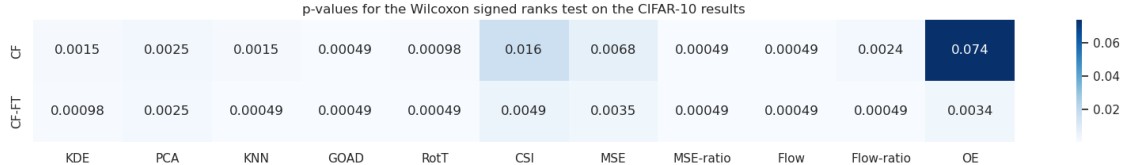

Figure 12: p-values for null-hypothesis "our model does not outperform the benchmark method" for all methods on CIFAR-10

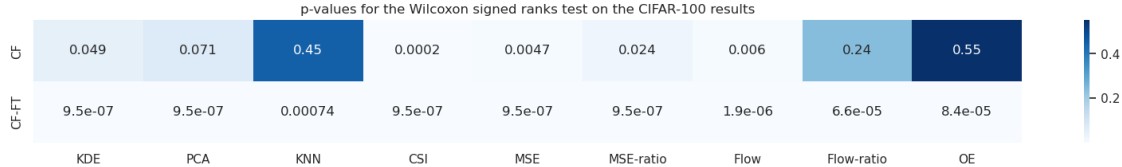

Figure 13: p-values for null-hypothesis "our model does not outperform the benchmark method" for all methods on CIFAR-100

all baseline methods significantly for a significance level of 0.05. For CelebA and the dataset setting, the number of samples (n=2 and n=3) is too small for the wilcoxon test. However, in the dataset-vs-dataset setting, our methods outperform the baselines by a clear margin of more than two standard deviations.

### A.7 Compute resources

Our experiments were performed on a GPU cluster using one cluster node with one Nvidia V100-32GB GPU and 2 CPU using 16 GB RAM. A training run of our contrastive normalizing flow took approximately 30 minutes. In this publication we show the results of about 200 training runs of the contrastive normalizing flow. Not included in this estimate are baseline method runs and research runs. Our organization is carbon neutral so that all its activities including research activities do not leave a carbon footprint. This includes our GPU clusters on which the experiments have been performed.

### A.8 Beyond the Image Domain

To further investigate our method we apply the contrastive normalizing flow to another data modality: We use the credit card fraud dataset from `https://www.kaggle.com/datasets/mlg-ulb/creditcardfraud`, and apply our contrastive normalizing flow. As a contrastive distribution we use the uncorrelated marginal distributions by permuting all samples in a batch per feature dimension. We show the results in figure 14. The contrastive normalizing flow and the standard normalizing flow perform on par (AUROC of 97.5), where the ratio method fails for outlier detection (AUROC of 41). Our contrastive normalizing flow does not outperform the standard normalizing flow, which is not surprising: Kirichenko et al. (2020) show that low-dimensional image features dominate the likelihood of the normalizing flow and thus lead to problems on image data. This behaviour is not expected on tabular data, where the standard normalizing flow is already a good outlier detection method. We think that the contrastive normalizing flow could be useful for other applications in the tabular and other domains: As shown in our ablation study in 5.2.3, the contrastive normalizing flow can be used for outlier detection with known anomalies.

### A.8.1 Discussion

For non-image settings, where no broad contrastive dataset like IMAGENET is readily available, the best choice is a set of known outliers. If necessary, this outlier dataset can be enlarged by data augmentation. If no outliers are known at training time, a potential fallback is to construct the contrastive distribution as a transformed inlier distribution, e.g., by adding Gaussian noise or using the product of the marginal distributions removing all correlations. However, we found in the tabular data experiment in section A.8

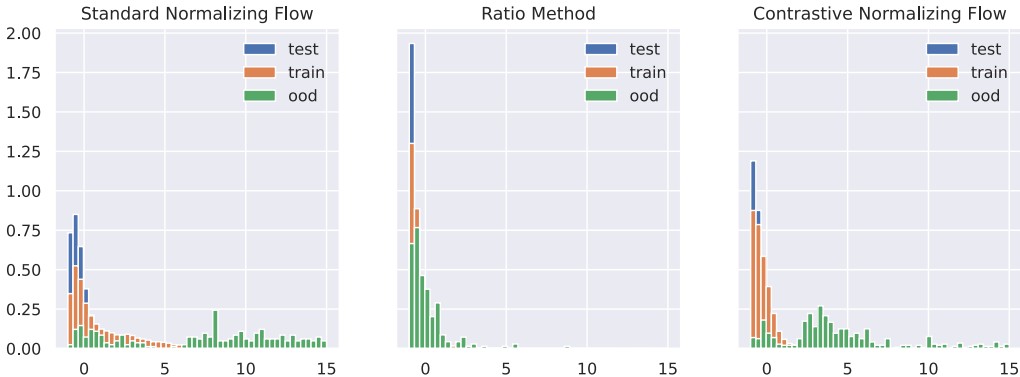

Figure 14: Standard Normalizing Flow, Ratio method and Contrastive Normalizing Flow applied on a tabular fraud detection dataset. While the ratio method completely fails in the tabular domain, the contrastive normalizing flow performs on par with the standard normalizing flow. We think that the contrastive normalizing flow can be used beyond the scope of image outlier detection in various data domains.

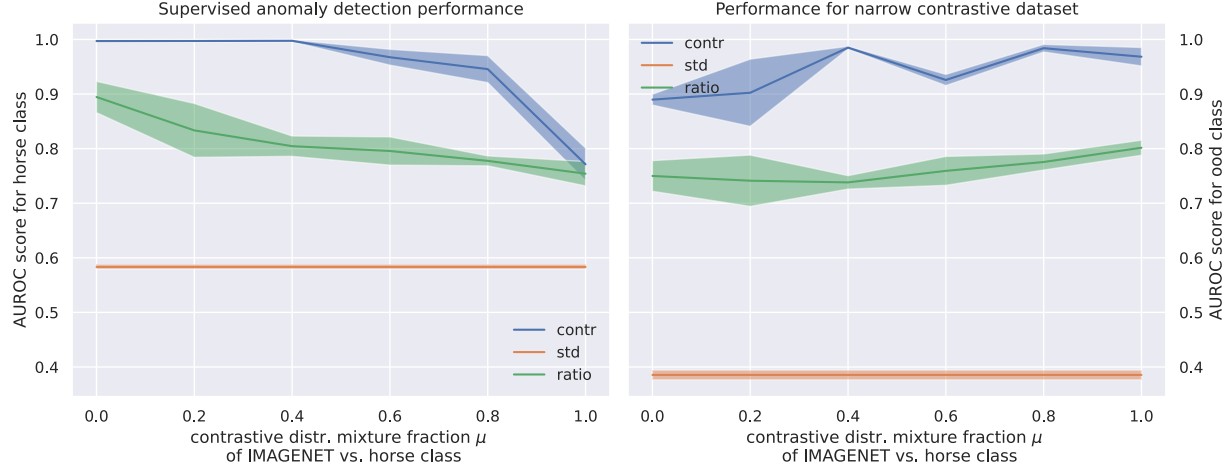

Figure 15: Training directly on images without a feature extractor: Performance for our model, the ratio method and a standard normalizing flow trained on the CIFAR-10 deer class as in-distribution and a mixture of IMAGENET and the horse class of CIFAR-10 as contrastive dataset. On the left we plot the AUROC score against the horse class included in the mixture (which correspond to a fully supervised setting when $\mu = 0.0$), on the right against all other CIFAR-10 classes (without horse and deer). Standard deviations are computed over three different seeds. Please note that although both graphs use the same trained model, the experimental setting is vastly different: On the left the distribution of the outliers is already known at training time, where on the right we stay in the uniformed outlier detection setting.

that this did not improve performance compared to a standard normalizing flow. We leave the choice of the contrastive distribution in the challenging setting without known outliers or broad contrastive distribution for future work.

## A.9 Applying contrastive normalizing flows on images without a feature extractor

In this section, we applied the contrastive normalizing flow directly on image data, utilizing the GLOW pytorch implementation from `https://github.com/rosinality/glow-pytorch` with the same experimental setup as our ablation studies in section 5.2. Due to longer compute times, we only conducted experiments on

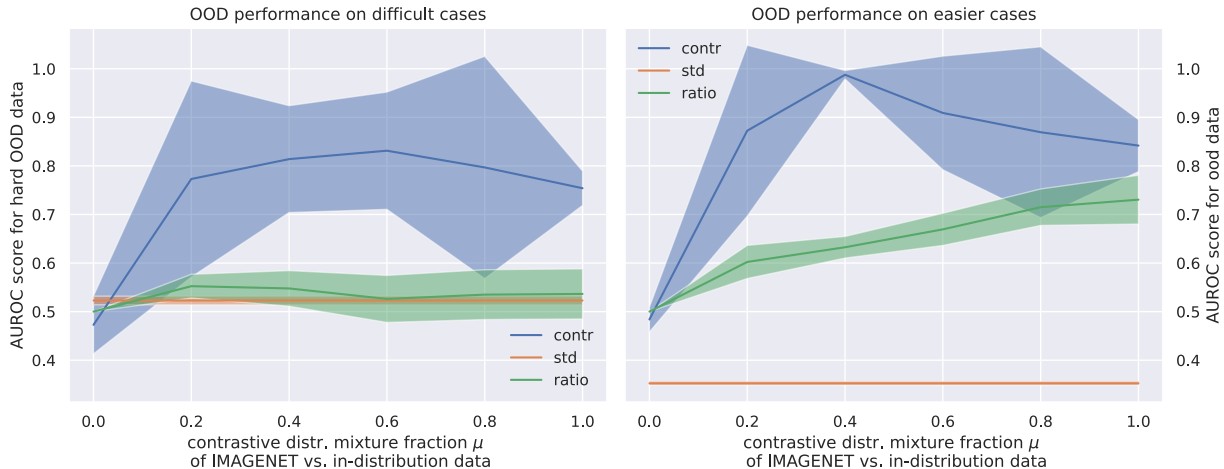

Figure 16: Training directly on images without a feature extractor: Performance for our model, a standard normalizing flow and the ratio method trained on the CIFAR-10 horse class as in-distribution and of a mixture of IMAGENET and in-distribution data as contrastive dataset. $\mu$ denotes the fraction of imagenet samples in the contrastive dataset, so for $\mu = 1$ the contrastive dataset consists only of imagenet samples and for $\mu = 0$ the contrastive dataset contains only samples from the CIFAR-10 horse class. On the right we plot the AUROC score against difficult out-of-distribution cases (deer class of CIFAR-10), on the left against all other CIFAR-10 classes (without horse and deer). Errors are computed over three different seeds.

a lower resolution of $\mu$ (in steps of 0.2). The results shown in Figures 15 and 16 support our findings in section 5.2 using a MoCo feature extractor: The contrastive normalizing flow reliably outperforms the ratio method and the standard normalizing flow. In the informed case, the performance of the contrastive flow without use of a feature extractor is superior to using MoCo, which supports our finding in the paper that the MoCo features have some overlap for semantically similar classes. In the uninformed cases, the performance of all three methods drops compared to the use of the MoCo feature extractor. As we are primarily interested in the uninformed outlier detection setting, the results confirm our choice of the MoCo feature extractor. This performance improvement can also be observed for standard outlier detection methods reported in table 8.

However, there are still several open questions and areas for improvement when employing contrastive normalizing flow on image data, e.g., sampling from the difference distribution trained on full images produces NaNs in the backward pass, which still needs to be investigated. It is possible that the training process requires further refinement and hyperparameter tuning, or we need a better understanding of high-dimensional (difference) distributions. Additionally, the contrastive normalizing flow could benefit from architectural improvements, as we observed that training using GLOW architecture on full images was unstable in comparison to training on MoCo features. In conclusion, further research is necessary to effectively apply contrastive normalizing flows directly on image data for outlier detection and other applications.

### A.10 Qualitative examples for inlier and anomalies

We show additional visual examples from the IN-DIST and OOD test set. IN-DIST test images with a low outlier score are shown in figure 17, IN-DIST test images with a high outlier score in figure 18 and OOD test images with a low outlier score shown in figure 19.

### A.11 Receiver Operating Characteristic

To further illustrate our method, we investigate ROC curves for the CIFAR-10 deer class. We use the same experimental setting as in figure 1: the deer class of CIFAR-10 is used as inlier distribution and IMAGENET is used for all contrastive methods as contrastive dataset. To see the performance for two corner cases, we

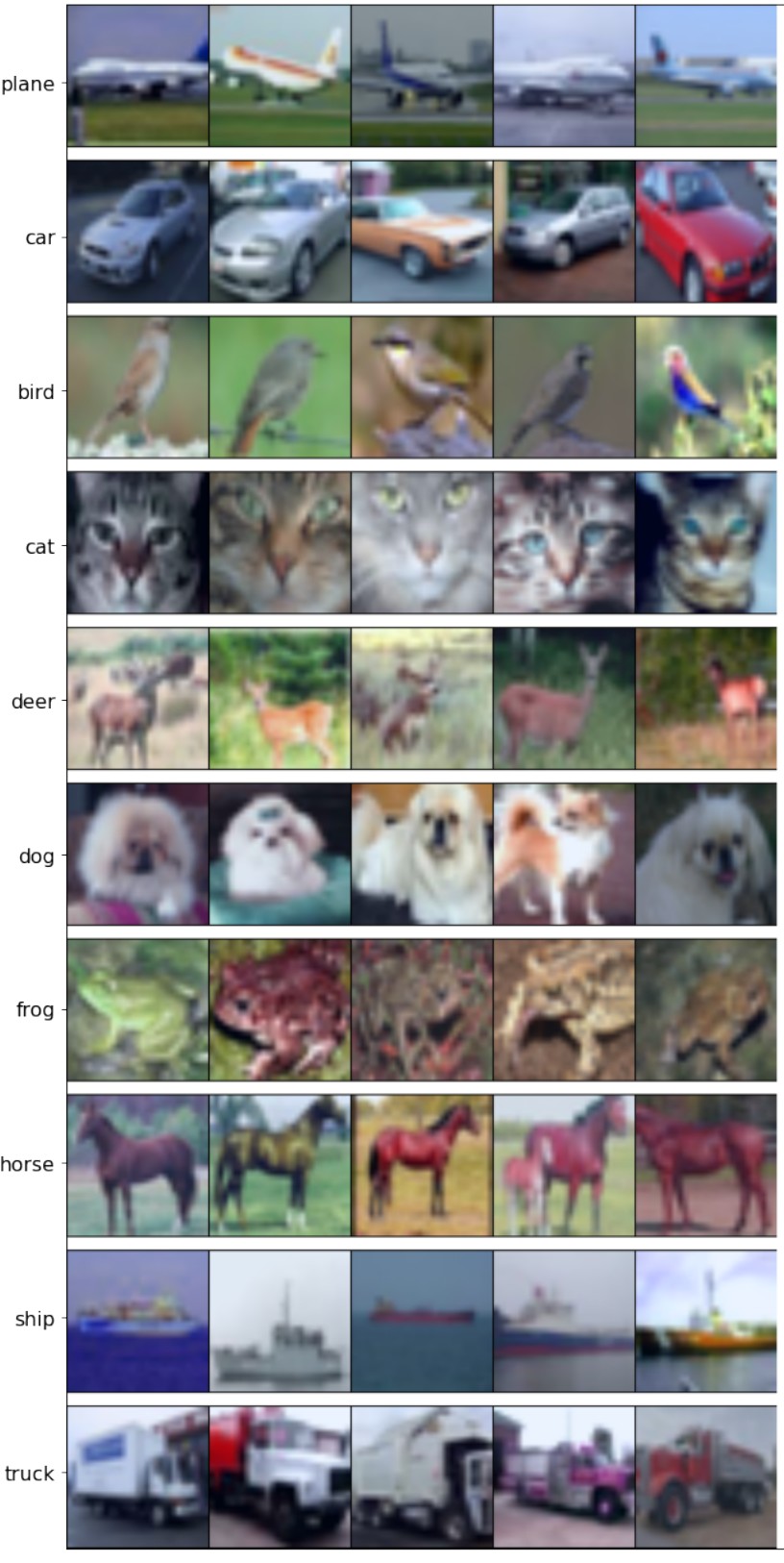

Figure 17: Examples of images with low outlier score of the IN-DIST test set for all CIFAR-10 classes. Every row shows examples for a model trained on one CIFAR-10 class as IN-DIST.

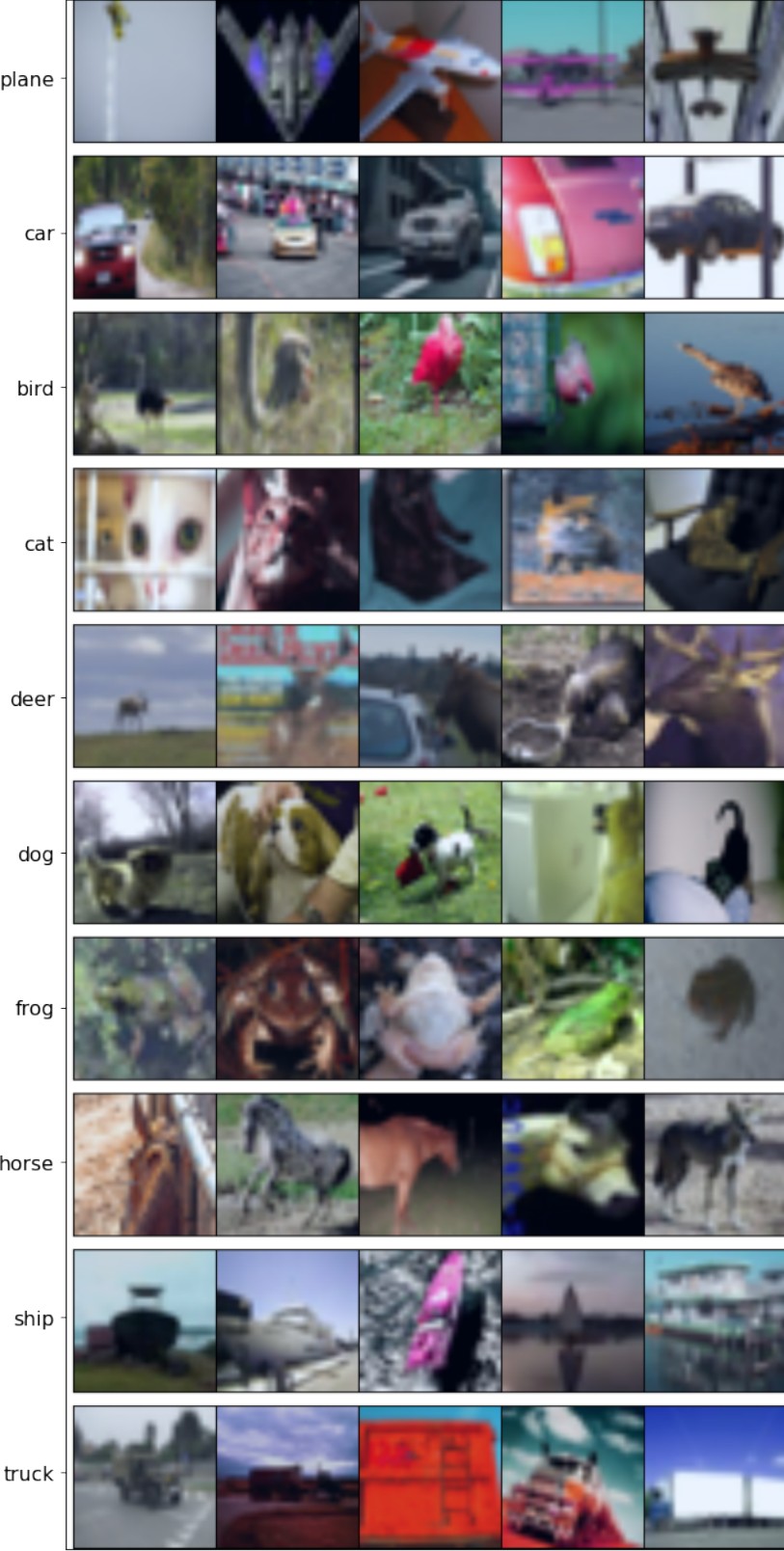

Figure 18: Examples of images with high outlier score of the IN-DIST test set for all CIFAR-10 classes. Every row shows examples for a model trained on one CIFAR-10 class as IN-DIST.

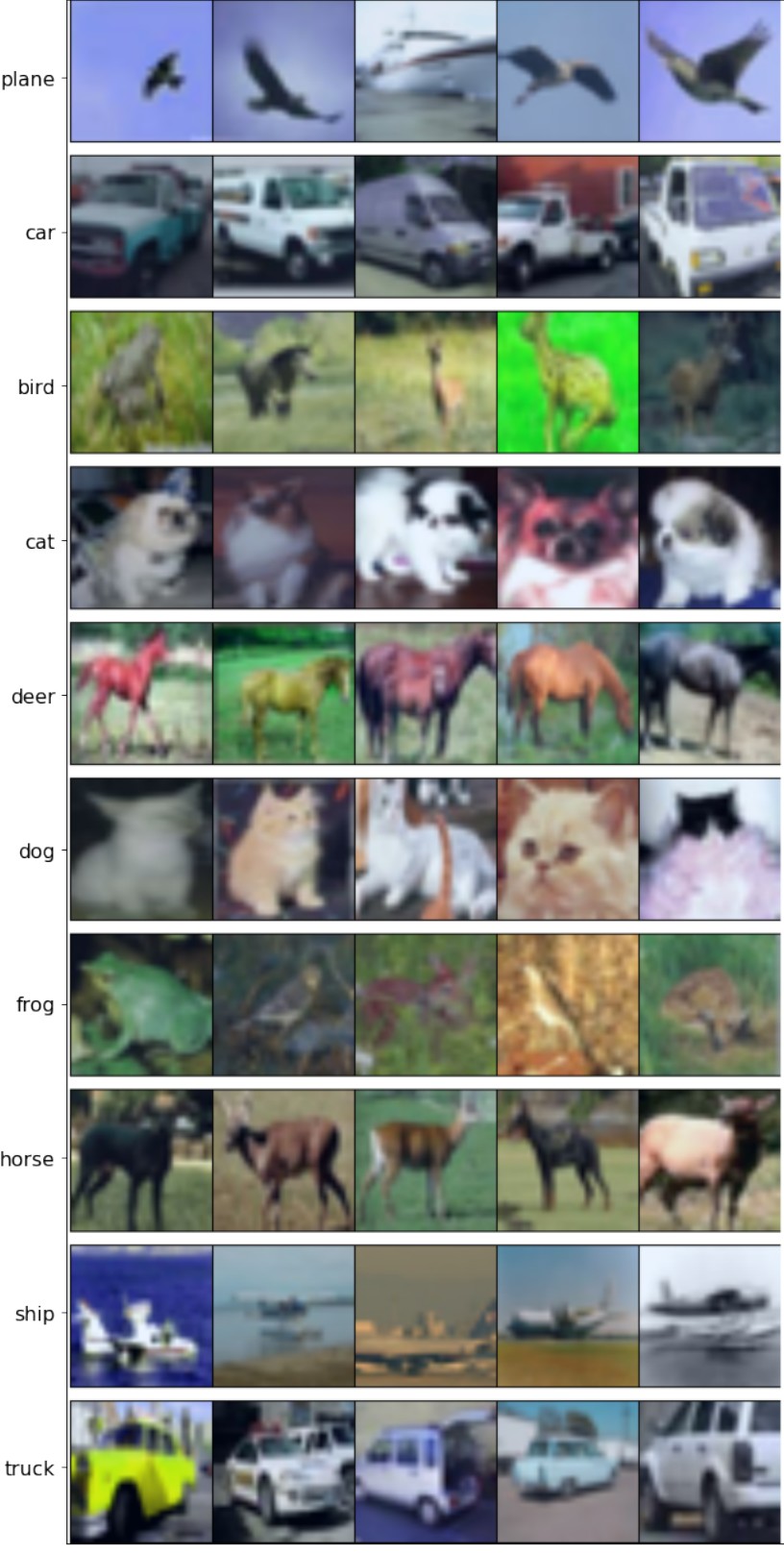

Figure 19: Examples of images with low outlier score of the OOD test set for all CIFAR-10 classes. Every row shows examples for a model trained on one CIFAR-10 class as IN-DIST.

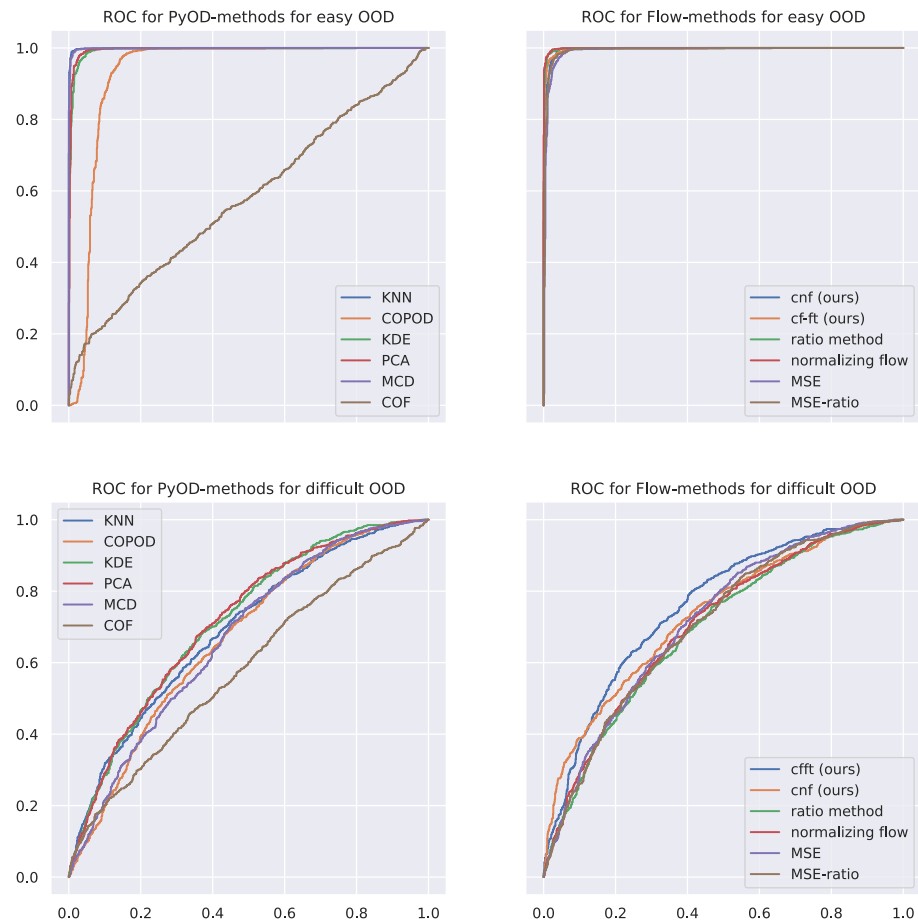

Figure 20: ROC curves for the CIFAR-10 deer class as in-distribution, IMAGENET as contrastive dataset and the horse class as difficult OOD cases and the truck class as easy OOD cases. We plot MSE and MSE-ratio with the flow models. We see that our contrastive normalizing flow method has an advantage over all other methods for low outlier scores in the case of difficult OOD samples. The fine-tuned contrastive normalizing flow improves for allmost all scores.

use two CIFAR-10 classes separately as anomalies: Our model showed for the pair of deer and horse class the worst performance over all CIFAR-10 pairs (when trained on deer as inlier), even in the supervises case (section 5.2.3) our model and the likelihood ratio model reach only an AUROC of 95. In contrast, for the deer class as inlier and truck class as outlier, our model shows almost perfect separation. See table 3 for the confusion matrix over all classes. The ROC curves for both outlier classes and various methods are shown in figure 20. When plotting the outlier detection performance for the horse class of CIFAR-10 (hard outliers), our contrastive normalizing flow (cnf) and the fine-tuned normalizing flow (cf-ft) show the best performance with an AUROC of 73 respectively 75. For the truck class (easy outliers) almost all models can separate nearly perfectly between inliers and anomalies.

