# OpenReview forum: "Positive Difference Distribution for Image Outlier Detection using Normalizing Flows and Contrastive Data"
_TMLR — Accepted by TMLR_

### Review · Reviewer_GvHC · 2023-03-14

**Summary Of Contributions:**

This work proposes contrastive normalising flow (CF) for the task of outlier detection.  CF is a method that learns to model the difference of two densities, $p(x)$, (the inlier distribution) and $q(x)$ (the contrastive / negative sample distribution) on areas where $p(x) > q(x)$. By using the log probability under the learned density model as an outlier score, outliers occur when a) samples have higher density in $q(x)$ rather than $p(x)$, i.e., they come with high probability from the contrastive distribution and b) samples have low probability in both $p(x)$ and $q(x)$, i.e., they are overall unknown. The authors prove that learning this positive difference distribution amounts to training a density model on a simple objective that maximises the log probability of samples coming from $p(x)$ while minimising the log probability of samples coming from $q(x)$. As the method critically relies on evaluations of the log probabilities of samples, the authors employ a normalising flow for the density model which permits exact evaluation of the log probabilities. Furthermore, it should be mentioned that the authors perform density estimation on the feature space of a pre-trained feature extractor, rather the original input space directly.

Besides CF, the authors also propose CF-FT, where the normalising flow is firstly trained to maximise the log probability of samples from $p(x)$ and then a short fine-tuning procedure is done with the CF objective to learn to subtract the appropriate density from $p(x)$ according to $q(x)$.

Both methods are evaluated in a quite extensively on a variety of settings. Firstly, the authors show on toy tasks that CF is learning the correct difference density and how its differences with more traditional ratio based approaches manifest. Secondly, the authors experiment on outlier detection using a wide variety of inlier, $p(x)$ and outlier / contrastive, $q(x)$, distributions and show how different choices for these affect the outlier detection performance. Finally, they also perform several benchmark tasks where it is shown that both CF and CF-TF perform favourably against the current state-of-the-art.

**Audience:**

Yes

**Broader Impact Concerns:**

No concerns.

**Claims And Evidence:**

Yes

**Requested Changes:**

This work is generally solid and I am willing to recommend acceptance. I only have a few comments that can potentially further improve it (based on the aforementioned weaknesses):

a) It would be nice if the authors can show performance when training the densities directly on the input space. This would disentangle a bit the dependence on the pretrained feature extractor.

b) In the CF type of approaches the density is trained under a specific lower bound to the log-likelihood (the $\varepsilon$ parameter). It is unclear whether such a lower bound would also help the baselines, e.g., the single normalizing flow for $p(x)$ or the ratio based method of $p(x)/q(x)$.

Other than that,  there are a few typos / places where the explanation can be improved

- This might be personal preference, but “outlier score” is a bit counterintuitive for the $\log p(x)$ scores; you want a low $\log p(x)$ score to identify an outlier, so $\log p(x)$ is more of an inlier score. Alternative, $-\log p(x)$ could be treated as an outlier score.
- Figure 1, CF plot; you need to mention the $p(x) > q(x)$, as otherwise the term inside the logarithm can be negative.
- Page 4, section 3.1: it is mentioned that that NF maps the data to a normal distribution. While this is usually the case in practice, in theory it can be an arbitrary base distribution, so I would rephrase it to be more general.
- Eq. 1 is incomplete, as for equality one also needs the constant terms of the standard normal base density. It would also be better if the notation is changed to $T_\theta()$ to show the explicit dependence of $T$ on $\theta$.
- Eq. 2, the temperature $\tau$ needs to be inside the exponent on the enumerator and a parenthesis is needed in the denominator. If $\tau$ is outside the exponent it cancels out.
- Underneath Eq. 2 it is mentioned that infoNCE maximises the mutual information; technically,  it maximises a lower bound to the mutual information and it is unclear how tight this bound is.
- Page 6: a $\hat{\theta}$ is introduced on the second paragraph after Eq. 6; it is unclear what it is.
- $p(x) - q(x) = p(f|s)(p(s) - q(s))$: there is a mismatch between the r.h.s. and the l.h.s. as $x$ is not present. It would be better if it is written as $p(x) - q(x) = \int p(x|f, s) p(f|s)(p(s) - q(s))df ds$.
- Third line from the end of page 6: contrastive is mentioned twice in sequence.
- Page 11, third line of the “How to choose a good contrastive dataset” paragraph: ninty -> ninety.
- Page 14, first sentence after Table 2: by -> be.


**Strengths And Weaknesses:**

*Strengths*
1. Simple method that works well in practice
2. Clear exposition of the CF based methods, with intuitive explanations on why they might be better than more traditional distribution ratios
3. Quite extensive experiments that provide adequate evidence for the claims and performance of the methods

*Weaknesses*
1. The method relies on a pre-trained feature extractor and results can degrade depending on its choice (c.f. Table 10 at the appendix)
2. Optimization is fundamentally unstable and the authors are clamping the log probabilities during training to avoid the instabilities. The amount of clamping is an additional hyperparameter that needs to be tuned. Similar clamping can also be applied to the baselines, (to e.g., avoid the instabilities when doing the ratio $p(x)/q(x)$), when e.g., one is learning $\log p(x)$ and $\log q(x)$.

---

> ### Author Response · Authors · 2023-03-31
> **Answer to reviewer GvHC**
>
> We express our gratitude to reviewer GvHC for the thorough review and valuable feedback provided.
>
> **Training directly on images**
>
> We conducted additional experiments in the setting of our ablation studies (section 4.2) where we directly trained on images using a glow normalizing flow and report the results in appendix A.9 of the revised version. The results support our findings using a MoCo feature extractor: The contrastive normalizing flow reliably outperforms the ratio method and the standard normalizing flow. In the informed case, the performance of CF without use of a feature extractor is superior to using MoCo, which supports our finding in the paper that the MoCo features have some overlap for semantically similar classes. In the uninformed cases, the performance of all three methods drops compared to the use of the MoCo feature extractor. As we are primarily interested in the uninformed outlier detection setting, the results confirm our choice of the MoCo feature extractor. This performance improvement can also be observed for standard outlier detection methods reported in table 8 in the appendix. Further research is needed to improve the training directly on images and reduce the difficulties of high-dimensional density estimation.
>
> **Epsilon clamping**
>
> We use the threshold epsilon to define a bound on the likelihood range of the contrastive data of which the likelihood is pulled down. For the inlier data, of which the likelihood is maximized, we do not use any threshold. Any contrastive sample with a log likelihood below -epsilon is considered "bad enough," meaning that there is no need to further decrease its likelihood. However, for standard normalizing flow training, such as with the standard flow and ratio method, one aims to maximize the log likelihood of all samples during training. It is unclear how to introduce a meaningful threshold during training. When we tried clamping the likelihoods of the standard and ratio normalizing flows during training, we did not observe any improvement but rather a deterioration of the outlier detection performance.
>
> **Typos / further improvements**
>
> We changed all mentioned passages in the revised version and thank the reviewer for the suggestions.

---

> > ### Comment · Reviewer_GvHC · 2023-04-07
> > **Response to authors**
> >
> > I would like to thank the authors for their response and manuscript updates, which address all of my concerns.

---

### Review · Reviewer_ifF9 · 2023-03-25

**Summary Of Contributions:**

This paper proposes an alternative objective for learning a flow model based on a target in-distribution dataset and a contrastive dataset.
In contrast to likelihood ratios, this method learns a positive difference distribution (where the difference is w.r.t. to the PDFs).
This method seems to be more stable than likelihood ratios or simple likelihood, and is experimentally shown to be competitive or outperform standard baselines.
More broadly, the training of a positive difference distribution may be interesting, though the paper does not discuss this.



**Audience:**

Yes

**Broader Impact Concerns:**

No broader impacts section but it does not seem required for this work.


**Claims And Evidence:**

Yes

**Requested Changes:**

- I suggest changing the title from "based" to "for". It makes more sense as "based" might require hyphens like "model-based", i.e., "Positive Difference Distribution for Image Outlier Detection using Normalizing Flows and Contrastive Data".


**Strengths And Weaknesses:**

I reviewed the prior version of this work so my comments only address the main changes. Overall, the additions and explanations improved the paper significantly.

*Comments on updates*
- The fundamental idea that ratios are unstable when learned from samples---particularly when outliers of both the original and contrastive distributions---compared to differences is very important. I would suggest highlighting this idea more near the end of section 4. Perhaps provide an enumerated list of the key benefits and distinctions from likelihood ratio or likelihood.

- I appreciated the explanation and logical flow of why use a feature extractor in this case.

- I also appreciated the comparison to other feature extractors and at least some justification for MoCo, both in words and in new experiments.

*Questions*
- Why are there two modes in Figure 1 (right)? Is there a good explanation for this?

- Is the positive difference distribution useful for anything beyond outlier detection? At this point, it only seems like outlier detection, but could it be useful more broadly?


*Minor Comments*
- You should point out the $\mu = 0$ case earlier and why normalizing flows work better in this case. Also, the use of $\mu$ is somewhat confusing. To double check, does $\mu=1$ mean that all the contrastive examples are from ImageNet? Perhaps clarifying the wording could help as it took me several reads to understand the figure.  Perhaps just something like "% Contrastive that is in-distribution".

- "semantic distant classes" -> "semantically distant classes"

---

> ### Author Response · Authors · 2023-03-31
> **Answer to Reviewer ifF9**
>
> We thank Reviewer ifF9 for taking the time and reviewing our paper for the second time. We are pleased to hear that the changes have significantly improved the paper.
>
> **Highlight advantages of the contrastive flow compared to ratio and standard nf**
>
> Following the suggestion, we added an overview table at the end of section 4 to highlight advantages and distinctions of our method in comparison to the ratio method and the standard normalizing flow.
>
> **Two modes in figure 1**
>
> One can understand the two modes as corresponding to the two terms of equation (5): Specifically, the first mode represents samples with p>q, while the second mode mode comprises samples with p<q. Since the likelihood of the contrastive data gets pulled down to the threshold epsilon, the second is located at zero in figure 1 (the threshold chosen for this experiment). In Figure 10 in the appendix, the relation between the second mode and the threshold epsilon is better observed.
>
> **Applications beyond outlier detection**
>
> We see multiple potential applications beyond outlier detection, e.g., for dataset reduction (e.g., training on a large unlabeled dataset and removing unwanted labeled data), but we think further research is needed to better understand high-dimensional difference distributions.
>
> **$\mu$**
>
> As the reviewer correctly states, for $\mu$=1 all contrastive samples stem from IMAGENET. We added the description of the extreme cases of $\mu$ = {0,1} in the revised version for clarification.
>
> **Title change**
>
> Lastly, we thank the reviewer for suggesting a title change, and we have modified the title accordingly.

---

### Comment · Reviewer_DjDq · 2023-03-06
**Thanks for your careful response**

thanks for the significant update and revision. I think the authors well address my concerns and questions. So I recommend to accept this work for publication.

---

> ### Author Response · Authors · 2023-03-31
> **Answer to Reviewer DjDq**
>
> We are glad to hear that the revision well addressed your concerns and questions. We thank you for your recommendation to accept the work for publication.

---

### Author Response · Authors · 2023-03-31
**Revised Version**

We uploaded the revised version of our paper, changes to the submission are marked in blue, all changes to the first submission are now marked in grey. We thank all reviewers for their positive and valuable feedback!

---

### Decision · Action_Editors · 2023-04-22

**Recommendation:** Accept as is

**Comment:**

The paper received two positive reviews:

1. Reviewer ifF9, who also reviewed the previous version of the paper, is satisfied with the improvements in this version.
2. Reviewer GvHC, who reviewed the paper for the first time, is satisfied with the clarity of the exposition and the experimental evaluation.

Reviewer DjDq, who reviewed the previous version, did not submit a review, but stated that they are happy to recommend the paper for acceptance. The editors-in-chief and I agreed that this recommendation is sufficient, so no further review was requested.

Since all reviewers recommend acceptance without further concerns, and since the paper has already undergone multiple revisions (including a major one), I'm happy to recommend acceptance as is.

**Audience:**

The paper is clearly of interest to TMLR's audience.

In the previous version of the submission, I wrote:

> reviewers expressed concerns about the paper's organization and clarity, which could potentially hamper the paper's reach to TMLR's audience

As the reviewers have noted, this version has made significant improvements in organization and clarity, so this is no longer a concern.

**Claims And Evidence:**

In the previous version of this submission, I wrote:

>the paper should either revise the claim about the method being state-of-the-art, or provide stronger empirical evidence to support the claim

The new version has provided additional empirical evidence to support the claim. All reviewers are satisfied that the claims made are adequately supported.